# Diagonal Symmetrization of Neural Network Solvers for the Many-Electron Schrödinger Equation

**Kevin Han Huang** [1]  **Ni Zhan** [2]  **Elif Ertekin** [3]  **Peter Orbanz** [1]  **Ryan P. Adams** [2]

## Abstract

Incorporating group symmetries into neural networks has been a cornerstone of success in many AI-for-science applications. Diagonal groups of isometries, which describe the invariance under a simultaneous movement of multiple objects, arise naturally in many-body quantum problems. Despite their importance, diagonal groups have received relatively little attention, as they lack a natural choice of invariant maps except in special cases. We study different ways of incorporating diagonal invariance in neural network ansätze trained via variational Monte Carlo methods, and consider specifically data augmentation, group averaging and canonicalization. We show that, contrary to standard ML setups, in-training symmetrization destabilizes training and can lead to worse performance. Our theoretical and numerical results indicate that this unexpected behavior may arise from a unique computational-statistical tradeoff not found in standard ML analyses of symmetrization. Meanwhile, we demonstrate that post hoc averaging is less sensitive to such tradeoffs and emerges as a simple, flexible and effective method for improving neural network solvers.

## 1. Introduction

We study the effect of symmetrizing neural network solutions to the Schrödinger equation. Solving the many-body Schrödinger equation is of fundamental importance in science, because it provides the key to understanding and predicting the behavior of quantum systems and thereby many

---

[1]Gatsby Unit, University College London, London, UK [2]Department of Computer Science, Princeton University, Princeton, NJ, USA [3]Department of Mechanical Science and Engineering, University of Illinois at Urbana-Champaign, Champaign, IL, USA. Correspondence to: Kevin Huang <han.huang.20@ucl.ac.uk>.

*Proceedings of the $42^{nd}$ International Conference on Machine Learning*, Vancouver, Canada. PMLR 267, 2025. Copyright 2025 by the author(s).

physical phenomena. Ab initio computational methods seek to solve the non-relativistic electronic Schrödinger equation from first principles. There, the computation is performed directly from physical constraints and without relying on empirical approximations or training data, with the promise of producing high-accuracy electronic wavefunctions. However, the strict requirement on physical constraints makes it challenging to incorporate neural networks into these methods. Carleo & Troyer (2017), Hermann et al. (2020) and FermiNet (Pfau et al., 2020) are some of the first successful ab initio neural network methods, which learn the ground state wavefunctions in atoms and molecules via a variational Monte Carlo (VMC) approach. Many neural network methods have emerged since then (Li et al., 2022; von Glehn et al., 2023; Cassella et al., 2023), each seeking to improve how physical constraints are modelled in different systems. While these approaches have produced state-of-the-arts results on ground state energy and other physical properties, one notable drawback is their exorbitant training cost compared to classical VMC methods. This issue becomes dire for modelling large systems, as the Hilbert space of wavefunctions grows exponentially with the number of electrons.

In other AI-for-science approaches, symmetrization has proved to be successful both for modelling physical constraints and for improving neural network performance (Batatia et al., 2022; Du et al., 2022; Batzner et al., 2022; Duval et al., 2023). One VMC example is DeepSolid (Li et al., 2022), a FermiNet-type wavefunction that employs translationally invariant features to model periodic solids. However, recent findings in protein structures (Abramson et al., 2024), atomic potential (Qu & Krishnapriyan, 2024) and machine learning theory (Huang et al., 2022; Balestriero et al., 2022) demonstrate that symmetries may be unnecessary and could sometimes be harmful for performance. This raises the question how much the previously observed improvements can be attributed to symmetries versus other factors such as hyperparameter and architecture choices.

Motivated by these questions, we examine the effectiveness of symmetrizing ab initio neural network solvers under the natural symmetry groups of a many-body problem, which are the *diagonal groups of isometries*. These groups, roughly speaking, describe the invariance of the system under a

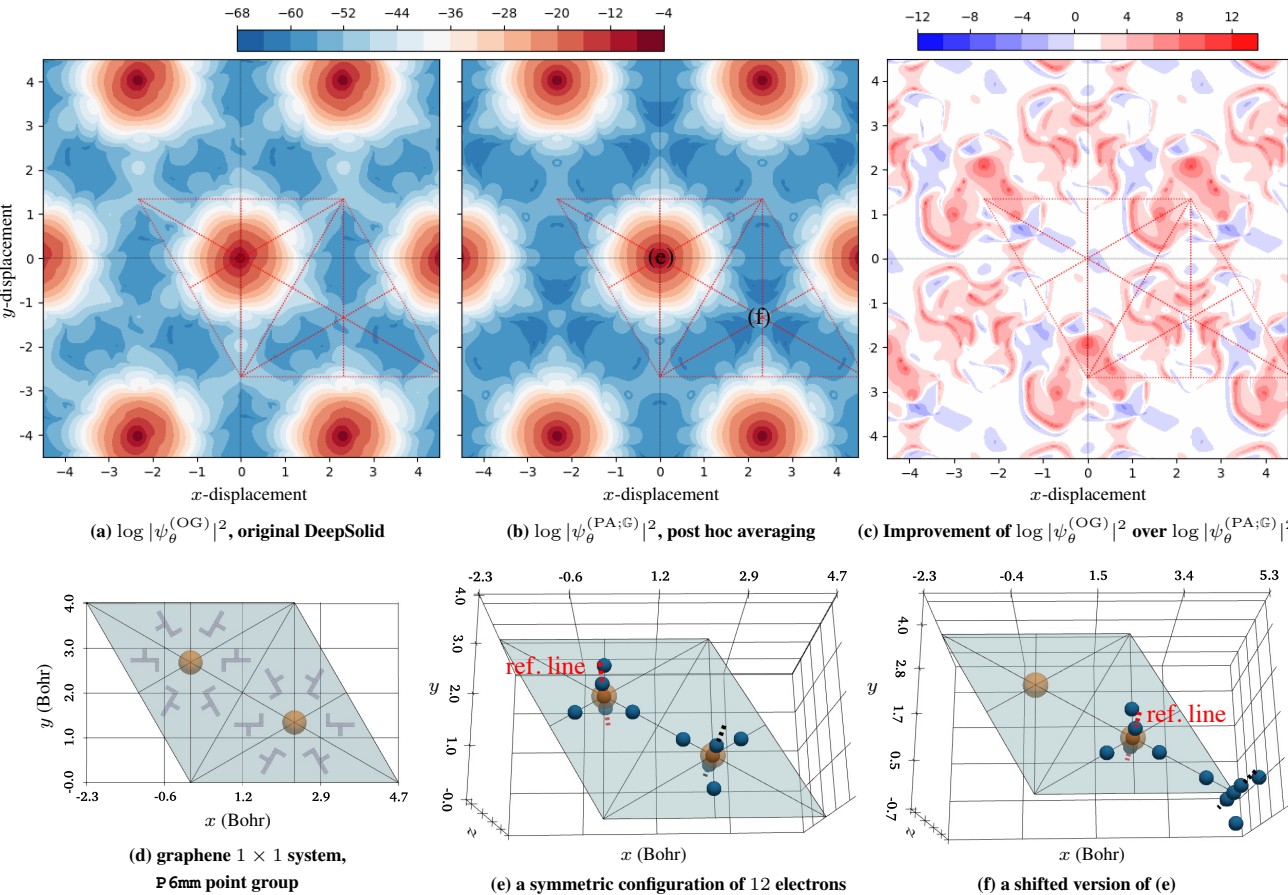

**(a)** $\log|\psi_\theta^{(\mathrm{OG})}|^2$, **original DeepSolid**      **(b)** $\log|\psi_\theta^{(\mathrm{PA};\mathbb{G})}|^2$, **post hoc averaging**      **(c)** **Improvement of** $\log|\psi_\theta^{(\mathrm{OG})}|^2$ **over** $\log|\psi_\theta^{(\mathrm{PA};\mathbb{G})}|^2$

**(d)** **graphene** $1 \times 1$ **system,**      **(e)** **a symmetric configuration of** $12$ **electrons**      **(f)** **a shifted version of (e)**
**$\mathbb{P}6mm$ point group**

*Figure 1.* Visualizations of the (partial) diagonal invariance of an unsymmetrized wavefunction versus a symmetrized wavefunction in a graphene $1 \times 1$ system. (a) and (b) are generated by evaluating $\log|\psi(x_1 + t, \ldots, x_{12} + t)|^2$ under a simultaneous 2d translation $t$ of the configuration $(x_1, \ldots, x_{12})$ given by the 12 blue spheres in (e). The red overlay indicates the unit cell in (e) such that the ref. line is exactly at the origin when $t = 0$. (d) shows 2 atoms (orange spheres) in an $1 \times 1$ planar supercell, with $\mathbb{G}$ illustrated by the ⌐ objects. (e) and (f) are shifted copies of the same configuration, with their positions marked in (b). This method only visualizes the partial $\mathbb{P}3m1$ symmetry; see Appendix C for a method that shows the full $\mathbb{P}6mm$ symmetry of $\psi_\theta^{(\mathrm{PA};\mathbb{G})}$. Details setups are in Sec. 6, 7 and Appendix B.

simultaneous movement of multiple objects; see Sec. 2 for a detailed review. Common symmetrization approaches in machine learning (ML) roughly fall under three categories:

- Randomly transforming data by symmetry operations (*data augmentation*);
- Averaging over group operations (*group averaging*);
- Invariant maps (*invariant features* and *canonicalization*).

We compare these approaches on DeepSolid (Li et al., 2022), the state-of-the-arts architecture for VMC on solids, for different solid systems. The emphasis is on providing an apples-to-apples comparison by fixing the architecture and hyperparameters, while varying the symmetry parameters. We find that, perhaps surprisingly, the effects of diagonal symmetrization for VMC problems are mixed and nuanced. This arises due to the unique combination of challenges posed by VMC and diagonal invariance, as well as the joint consideration of computational cost, statistical behaviors and physical constraints. In particular, our analyses indicate:

*In-training symmetrization can hurt.* VMC training operates

in an "infinite-data" regime, and every symmetry operation comes at the cost of forgoing one new data point. Holding the computational budget constant, symmetrization can destabilize training and lead to worse performance. This is demonstrated by theoretical and numerical results in Sec. 4.

*Post hoc symmetrization helps.* At inference time, VMC solvers are less sensitive to computational costs, and allow for averaging over a moderate number of group operations (Sec. 5). We show that post hoc averaging (PA) leads to improved energy, variance and symmetry properties of the learned wavefunction (Fig. 1, Table 2). In one case, post hoc averaged DeepSolid achieves performance close to Deep-Solid trained with $10\times$ more computational budget (Sec. 7).

The remainder of the paper provides mathematical discussions and computational tools for understanding diagonal symmetries in VMC. Sec. 2 reviews the concept of diagonal invariance and discusses why, except for simple cases e.g., translations or $E(3)$, finding a natural smooth invariant map is difficult. This is further corroborated by mathematical and

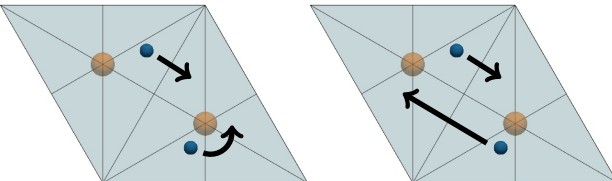

*Figure 2.* Different invariances for two electrons in a graphene system. *Left.* Separate invariance under a reflection and a rotation. *Right.* Diagonal invariance under a simultaneous reflection.

empirical results in the special case of a *smoothed canonicalization* (Sec. 4.3 and Appendix E). Sec. 3 briefly reviews the VMC setup and how different computational costs arise. Sec. 4 and 5 respectively examine in-training and post hoc symmetrization. Sec. 6 discusses our evaluation metrics and develops a method for visualizing diagonal symmetries (Fig. 1). Sec. 7 discusses experiment details. Additional results and proofs are included in the appendix.

### 1.1. Many-body Schrödinger equation

Throughout, we shall consider finding the $n$-electron ground state wavefunction $\psi : \mathbb{R}^{3n} \to \mathbb{C}$ and the corresponding minimal energy $E \in \mathbb{R}$ of the Schrödinger equation

$$H\psi(\mathbf{x}) = E\psi(\mathbf{x}), \quad \mathbf{x} := (x_1, \ldots, x_n) \in \mathbb{R}^{3n} . \quad (1)$$

The Hamiltonian $H$ is given by $H\psi(\mathbf{x}) := -\frac{1}{2}\Delta\psi(\mathbf{x}) + V(\mathbf{x})\psi(\mathbf{x})$, $\Delta$ is the Laplacian representing the kinetic energy and $V : \mathbb{R}^{3n} \to \mathbb{R}$ is the potential energy of the physical system. We also denote neural network ansätze by $\psi_\theta$, parametrized by some network weights $\theta \in \mathbb{R}^q$. Note that in general, the wavefunction depends on each electron via $(x_i, \sigma_i)$, where $\sigma_i \in \{\uparrow, \downarrow\}$ is the spin, and $\psi$ is required to be anti-symmetric with respect to permutations of $(x_i, \sigma_i)$. We focus on the case with fixed spins for simplicity, as is done in FermiNet and DeepSolid.

## 2. Diagonal invariance

Many physical systems of interest exhibit symmetry under some group $\mathbb{G}$ of isometries. $\mathbb{G}$ consists of maps of the form

$$x \mapsto Ax + b , \qquad x \in \mathbb{R}^3 , \quad (2)$$

for some orthogonal $A \in \mathbb{R}^{3\times3}$ and some translation $b \in \mathbb{R}^3$. We focus on groups that are countable. For systems with $n > 1$ electrons, $\mathbb{G}$ typically causes the potential $V$ in (1) to be invariant under a *diagonal group* $\mathbb{G}_{\text{diag}}$ acting on $\mathbb{R}^{3n}$:

$$\mathbb{G}_{\text{diag}} := \{(g, \ldots, g) \mid g \in \mathbb{G}\} .$$

To see how $\mathbb{G}_{\text{diag}}$ may arise, consider the Coulomb potential

$$V_{\text{Coul}}(\mathbf{x}) = \sum_{i<j} \frac{1}{\|x_i - x_j\|} + \sum_{i,I} \frac{1}{\|x_i - r_I\|} + \ldots ,$$

Each $r_I$ is the fixed, known position of the $I$-th atom (under the Born-Oppenheimer approximation), $\|\bullet\|$ is the Euclidean norm, and ... are the omitted electron-independent

terms. If the set of atom positions $\{r_I\}$ is invariant under some $\mathbb{G}$, $V_{\text{Coul}}$ is invariant under a *simultaneous* transformation of all $x_i$'s by the same $g \in \mathbb{G}$. Note however that $V_{\text{Coul}}$ does *not* satisfy *separate invariance*, i.e. invariance does not hold if $x_1$ and $x_2$ are transformed by different group elements, since $\frac{1}{\|g_1(x_1) - g_2(x_2)\|} \neq \frac{1}{\|x_1 - x_2\|}$ in general. Mathematically, separate invariance can be modelled by the product group $\mathbb{G}^n$ acting on $\mathbb{R}^{3n}$, and diagonal invariance arises as a specific subgroup of $\mathbb{G}^n$, i.e. $\mathbb{G}_{\text{diag}} \subseteq \mathbb{G}^n$. Fig. 2 illustrates the difference between $\mathbb{G}_{\text{diag}}$ and $\mathbb{G}^n$ in a 2-electron system: Under the given symmetry, a potential function is left unchanged when both electrons are reflected, but not when one is reflected and the other is rotated.

**$\mathbb{G}_{\text{diag}}$-invariant wavefunction.** Throughout this paper, we use the shorthand $g(\mathbf{x}) = (g(x_1), \ldots, g(x_n))$ for $g \in \mathbb{G}$, and focus on $\mathbb{G}_{\text{diag}}$-invariant potentials $V$, i.e.,

$$V(\mathbf{x}) = V(g(\mathbf{x})) \qquad \text{for all } g \in \mathbb{G} . \quad (3)$$

Invariance of $V$ does not imply the invariance of $\psi$: For a translation-invariant $V$ with one electron, Bloch (1929) proves that $\psi$ is only invariant up to a phase factor, and non-invariant solutions can occur when the ground state is degenerate (Tinkham, 2003). Nonetheless, an invariant solution can always be constructed from a linear combination of these states. The next result confirms this for the general case of $n \geq 1$ electrons and a diagonal group of isometries.

**Fact 2.1.** *Suppose $(\psi, E)$ solves (1) and $V$ is invariant under some $\mathbb{G}_{\text{diag}}$ induced by a group $\mathbb{G}$ of isometries. Then for any finite subset $\mathcal{G}$ of $\mathbb{G}$,*

$$\psi^{\mathcal{G}}(\mathbf{x}) := \frac{1}{|\mathcal{G}|} \sum_{g \in \mathcal{G}} \psi(g(\mathbf{x})) \quad (4)$$

*also solves (1) with respect to same energy $E$. In particular, if $\mathcal{G}$ is a subgroup, $\psi^{\mathcal{G}}$ is invariant under $\mathcal{G}$.*

Fact 2.1 motivates us to seek wavefunctions that respect the $\mathbb{G}_{\text{diag}}$-invariance of the system. Fig. 1(a) also shows that an unsymmetrized, well-trained wavefunction already attempts to achieve some extent of approximate invariance.

For more references on how $\mathbb{G}_{\text{diag}}$-invariance arises and plays an important role in modelling $\psi$, see Rajagopal et al. (1995); Zicovich-Wilson & Dovesi (1998).

**Challenges in modelling $\mathbb{G}_{\text{diag}}$-invariance.** Despite successes in modelling simple $n$-electron symmetries such as the Euclidean group $E(3)$ (Batzner et al., 2022) and translations (Whitehead et al., 2016), generalizing the approaches in those settings to general isometries can be challenging:

- Translation groups possess simple and well-understood symmetries. Common symmetrizations include periodic Fourier bases (Rajagopal et al., 1995) and projection via taking a modulus (Dym et al., 2024). The former trades off representation power against computational cost as the

number of bases used varies, whereas the latter suffers from discontinuity and requires smoothing (Sec. 4.3). Neither has been extended to a general $\mathbb{G}_{\text{diag}}$.

- $E(3)$ consists of all isometries in $\mathbb{R}^3$, and also admits simple invariant features (Batzner et al., 2022). However, the additional symmetries under $E(3)$ are undesirable when $\mathbb{G}$ is only a subgroup of $E(3)$, since the missing asymmetries represent a loss of information and therefore limit the representative power of the feature. Indeed, building an invariant map without losing information necessitates the availability of a maximal invariant (Lehmann et al., 1986), whereas building a maximal invariant for $\mathbb{G}_{\text{diag}}$ while respecting continuity constraints requires extending the mathematical theory of orbifolds (Ratcliffe, 1994; Adams & Orbanz, 2023) to $\mathbb{G}_{\text{diag}}$ of isometries. Such extensions are not known to the best of our knowledge.

In short, invariant maps are well-understood for groups with simple isometries and with many isometries, and the difficult cases are often found in the ones with restricted symmetries. This problem is particularly pronounced in the case of a diagonal group. Mathematically, the symmetries of $\mathbb{G}_{\text{diag}}$ are described by a fixed group $\mathbb{G}$ in $\mathbb{R}^3$, but act on a larger and larger space $\mathbb{R}^{3n}$ as the number of electrons $n$ grows. This means that $\mathbb{G}_{\text{diag}}$ admits substantially less structure than that of e.g. the product group $\mathbb{G}^n$.

**Space group $\mathbb{G}_{\text{sp}}$ for a crystal system.** A particular difficult case of $\mathbb{G}_{\text{diag}}$-invariance, in view of the above discussion, is the one induced by a space group $\mathbb{G}_{\text{sp}}$ in a crystal lattice. These groups are described by (2) with a finite number of orthogonal matrices $A$ and a countable number of translations $b$, which tile the $\mathbb{R}^3$ space with a finite unit volume called the *fundamental region*. Fig. 1(d) illustrates the P6mm point group in a graphene system, where the space is tiled by a triangular fundamental region. The study of space group is a fundamental subject of solid state physics, and we refer interested readers to Ashcroft & Mermin (1976).

Since $\mathbb{G}_{\text{sp}}$ is infinite, Fact 2.1 does not apply directly to $\mathcal{G} = \mathbb{G}_{\text{sp}}$. In practice however, to avoid computing an infinite crystal lattice, a common VMC practice is to restrict the system to a finite volume called the *supercell* (Esler et al., 2010; Kittel & McEuen, 2018). This effectively reduces the set of translations $b$ in (2) to a finite set and hence $\mathbb{G}_{\text{sp}}$ to a finite group, which allows Fact 2.1 to apply.

While our theoretical results apply to all finite groups of isometries, our numerical experiments focus on $\mathbb{G}_{\text{sp}}$. Each $\mathbb{G}_{\text{sp}}$ is denoted by standard shorthands e.g., P6mm, and an exhaustive list of $\mathbb{G}_{\text{sp}}$ can be found in Brock et al. (2016). We emphasize that our focus on the diagonal action of $\mathbb{G}_{\text{sp}}$, a difficult class of restricted symmetries found in crystals, sets our work apart from existing VMC works that have investigated translations, simple point symmetries in molecules, as well as the continuous symmetries in $SO(2)$, $SU(2)$ and

$E(3)$. A non-exhaustive list of those works can be found in Mahajan & Sharma (2019); Lin et al. (2023); Luo et al. (2023); Zhang et al. (2025) and further comparisons are discussed in Section 4.3.

**Symmetrization of many-body wavefunctions.** We shall focus on generic symmetrization techniques from ML that can be applied directly to many-electron wavefunctions. This is to be distinguished with classical techniques, e.g., Zicovich-Wilson & Dovesi (1998): There, symmetrization is performed by building invariance into single-electron features, which does not generalize well to state-of-the-art neural network solvers that are many-body by design.

## 3. Neural network VMC solver

We briefly review the VMC approach for training our ab initio neural network solvers. VMC seeks to solve the minimum eigenvalue problem of (1) via the optimization

$$\underset{\theta \in \mathbb{R}^q}{\operatorname{argmin}} \frac{\langle \psi_\theta, H\psi_\theta \rangle}{\langle \psi_\theta, \psi_\theta \rangle} = \underset{\theta \in \mathbb{R}^q}{\operatorname{argmin}} \mathbb{E}[E_{\text{local};\psi_\theta}(\mathbf{X})], \quad (5)$$

where $\mathbf{X} \sim p_{\psi_\theta}$ and $E_{\text{local};\psi_\theta}(\mathbf{x}) := H\psi_\theta(\mathbf{X})/\psi_\theta(\mathbf{X})$ is the local energy. $\langle f, g \rangle := \int f(\mathbf{x})^* g(\mathbf{x}) \, d\mathbf{x}$ is the complex inner product, and $p_\psi(\mathbf{x}) = \frac{|\psi(\mathbf{x})|^2}{\langle \psi, \psi \rangle}$ is the probability distribution obtained by normalizing a wavefunction $\psi$. The optimization may be performed by first or second-order methods, which can be represented by the generic functions $F_{\mathbf{x};\psi_\theta} \equiv F(\psi_\theta(\mathbf{x}), \Delta\psi_\theta(\mathbf{x})) \in \mathbb{R}^q$ and $Q_{\mathbf{x};\psi_\theta} \equiv Q(\psi_\theta(\mathbf{x}), \Delta\psi_\theta(\mathbf{x})) \in \mathbb{R}^{q \times q}$ as

$$\theta \mapsto \theta - \mathbb{E}[F_{\mathbf{X};\psi_\theta}], \quad \theta \mapsto \theta - \mathbb{E}[Q_{\mathbf{X};\psi_\theta}]^{-1}\mathbb{E}[F_{\mathbf{X};\psi_\theta}],$$

Examples of $F_{\mathbf{X};\psi_\theta}$ and $Q_{\mathbf{X};\psi_\theta}$ can be found in Pfau et al. (2020). Notably, the expectation formulation above converts the expensive integral over the entire space into an expectation, which can then be approximated by Monte Carlo averages computed on finitely many samples from $p_{\psi_\theta}$.

**Training (optimization) phase.** Every training step consists of two sub-steps: (i) *Sampling.* Samples are obtained from running $N$ independent MCMC chains with $p_{\psi_\theta}$ as the target distribution; (ii) *Gradient computation.* $F_{\mathbf{X};\psi_\theta}$ (or $Q_{\mathbf{X};\psi_\theta}$) is computed on the $N$ samples. Since (i) typically requires computing only the derivative $\partial_x \psi_\theta$, whereas (ii) involves at least $\partial_\theta \psi_\theta$ and $\partial_x^2 \psi_\theta$, the one-step computational costs of (i) and (ii) typically compare as $C_{\text{samp}} \ll C_{\text{grad}}$. This is particularly true for neural network solvers, where short chains ($20 - 100$ steps) are typically used due to the expensive gradient evaluation and that any small increase in per-step cost is amplified by the large number of training steps. We verify this cost comparison in Table 1 and discuss the alternative case with $C_{\text{samp}} \geq C_{\text{grad}}$ in Appendix I.

**Inference phase.** Having obtained a trained wavefunction $\psi_{\hat{\theta}}$ parametrized by $\hat{\theta}$, we draw samples from *long* chains

that target $p_{\psi_{\hat{\theta}}}$. These samples are used to compute various physical properties of $\psi_{\hat{\theta}}$ that can be expressed as expectations of $p_{\psi_{\hat{\theta}}}$; see Sec. 6 for details. Note that the only computational cost occurred here is in terms of $C_{\text{samp}}$.

Symmetrization can be performed during training, inference or both. We analyze symmetrization techniques in the two phases separately in Sec. 4 and Sec. 5, as they involve different computational tradeoffs. Our theoretical analysis considers first-order methods for simplicity. Our numerical results focus on the symmetrization of DeepSolid (Li et al., 2022), a state-of-the-arts neural network solver for solid systems with $\mathbb{G}_{\text{sp}}$-symmetry, trained with the second-order method KFAC (Martens & Grosse, 2015); see Appendix B.1.

# 4. Symmetrization during training

We first present a theoretical analysis of the behavior of gradient updates under different in-training symmetrization techniques. Let $\mathbf{X}_1, \ldots, \mathbf{X}_N$ be i.i.d. samples from $p_{\psi_\theta}^{(m)}$, the distribution of an $m$-th step MCMC chain with $p_{\psi_\theta}$ as the target distribution. Our benchmark for comparison is the update rule with the original (OG) unsymmetrized $\psi_\theta$,

$$\theta \mapsto \theta - \delta\theta^{(\text{OG})}, \quad \delta\theta^{(\text{OG})} := \frac{1}{N}\sum_{i \leq N} F_{\mathbf{X}_i;\psi_\theta}. \quad (6)$$

**Sample size bottlenecked by computational cost.** A key element of our analysis is that the VMC methods are in an *infinite data regime*. More precisely, unlike setups where sample size is constrained by the number of data points — commonly found in many theoretical analyses of symmetrization techniques (Chen et al., 2020; Lyle et al., 2020; Huang et al., 2022) — we are theoretically allowed to draw infinitely many samples from the sampling step during training. The practical limitation comes from $C_{\text{samp}}$ and $C_{\text{grad}}$, both of which are affected by the batch size $N$ as well as the symmetrization techniques used. By taking the computational effects into account, we shall see that symmetrization techniques exhibit substantially different statistical behaviors from those observed in the existing literature.

## 4.1. Pitfalls of data augmentation (DA)

Since "training data" correspond to samples drawn from $p_{\psi_\theta}^{(m)}$, a $k$-fold data augmentation is performed as follows:

(i) Sample $N/k$ $\mathbf{X}_1, \ldots, \mathbf{X}_{N/k} \overset{\text{i.i.d.}}{\sim} p_{\psi_\theta}^{(m)}$;
(ii) Sample $\mathbf{g}_{1,1}, \ldots, \mathbf{g}_{N/k,k}$ i.i.d. from some distribution on $\mathbb{G}$;
(iii) Compute the DA update as $\theta \mapsto \theta - \delta\theta^{(\text{DA})}$, where

$$\delta\theta^{(\text{DA})} := \frac{1}{N}\sum_{i \leq N/k}\sum_{j \leq k} F_{\mathbf{g}_{i,j}(\mathbf{X}_i);\psi_\theta}.$$

Notice that the sample size in (i) is reduced to $N/k$, since a set of $k$-times augmented $N/k$ samples and a set of unaug-

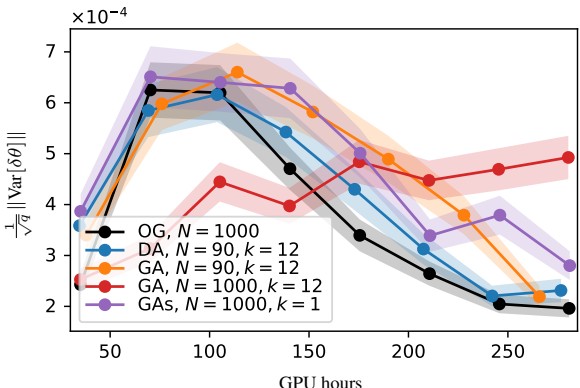

*Figure 3.* Normalized variance of differently symmetrized gradient updates against GPU hours. Experiment details in Sec. 7.

| Method | $N$ | $k$ | $C_{\text{samp}}$ (s) | $C_{\text{grad}}$ (s) | Total (s) |
|--------|-----|-----|----------|----------|-----------|
| OG | 1000 | - | 0.16(3) | 2.4(3) | 2.5(3) |
| DA | 90 | 12 | 0.041(5) | 2.4(2) | 2.5(2) |
| GA | 90 | 12 | 0.16(2) | 2.6(1) | 2.7(1) |
| GA | 1000 | 12 | 1.50(1) | 24(1) | 25(1) |
| GAs | 1000 | 1 | 0.16(1) | 2.4(1) | 2.5(1) |

*Table 1.* Computational cost per training step. Details in Sec. 7.

mented $N$ samples incur the same gradient evaluation cost $C_{\text{grad}}$, which dominates the overall computational cost.

While the sampling cost, $C_{\text{samp}}/k$, enjoys a minor speed-up, the next result shows that this comes at the cost of increased instability of the gradient estimate. Below, we denote the distribution of $\mathbf{X}_1^{\mathbf{g}} := \mathbf{g}_{1,1}(\mathbf{X}_1)$ by $p_{\psi_\theta;\text{DA}}^{(m)}$, and define a distance between two distributions $p, p'$ on $\mathbb{R}^{3n}$ as

$$d_{\mathcal{F}}(p, p') = \sup_{f \in \mathcal{F}} \|\mathbb{E}_p[f(\mathbf{X})] - \mathbb{E}_{p'}[f(\mathbf{X})]\|.$$

where $\mathcal{F}$ is a class of $\mathbb{R}^{3n} \to \mathbb{R}^{p+p^2}$ test functions such that

$$\left\{\mathbf{x} \mapsto \left(F_{\mathbf{x};\psi_\theta}, F_{\mathbf{x};\psi_\theta}^{\otimes 2}\right) \mid \theta \in \mathbb{R}^q\right\} \subseteq \mathcal{F}.$$

$d_{\mathcal{F}}$ is called an integral probability metric (Müller, 1997).

**Proposition 4.1.** *Fix $\theta \in \mathbb{R}^p$. Then*

$$\|\mathbb{E}[\delta\theta^{(\text{DA})}] - \mathbb{E}[\delta\theta^{(\text{OG})}]\| \leq d_{\mathcal{F}}\left(p_{\psi_\theta;\text{DA}}^{(m)}, p_{\psi_\theta}^{(m)}\right),$$

$$\left\|\text{Var}[\delta\theta^{(\text{DA})}] - \text{Var}[\delta\theta^{(\text{OG})}] - \frac{(k-1)\text{Var}\mathbb{E}[F_{\mathbf{X}_1^{\mathbf{g}};\psi_\theta}|\mathbf{X}_1]}{N}\right\|$$

$$\leq \frac{1 + 2\|\mathbb{E}[\delta\theta^{(\text{OG})}]\| + d_{\mathcal{F}}(p_{\psi_\theta;\text{DA}}^{(m)}, p_{\psi_\theta}^{(m)})}{N} d_{\mathcal{F}}\left(p_{\psi_\theta;\text{DA}}^{(m)}, p_{\psi_\theta}^{(m)}\right).$$

*In particular, if the distribution $p_{\psi_\theta}^{(m)}$ is invariant under $\mathbb{G}_{\text{diag}}$, i.e. $\mathbf{g}(\mathbf{X}_1) \overset{d}{=} \mathbf{X}_1$ for all $\mathbf{g} \in \mathbb{G}_{\text{diag}}$, we have*

$$\mathbb{E}[\delta\theta^{(\text{DA})}] = \mathbb{E}[\delta\theta^{(\text{OG})}] \quad \text{and} \quad \text{Var}[\delta\theta^{(\text{DA})}] \succeq \text{Var}[\delta\theta^{(\text{OG})}],$$

*where $\succeq$ is the Loewner order of non-negative matrices.*

The error $d_{\mathcal{F}}(p_{\psi_\theta;\text{DA}}^{(m)}, p_{\psi_\theta}^{(m)})$ describes how much an augmented sample from $m$-step chain deviates from an unaugmented sample on average, as measured through the gradient

$F_{x;\psi_\theta}$ and the squared gradient $F^{\otimes 2}_{x;\psi_\theta}$. For early training, we expect this error to have small contributions to the overall optimization compared to other sources of noise, e.g., the error incurred by running short chains instead of long chains and by stochastic gradients. At the final steps of training, we expect this error to be small as $p^{(m)}_{\psi_\theta}$ becomes approximately invariant; Fig. 1(a) shows that this is the case for an unsymmetrized, well-trained neural network.

Several messages follow from Proposition 4.1:

**DA leads to similar gradients in expectation but a possibly worse variance**. This is in stark contrast to known analyses of DA in the ML literature, where DA for empirical averages is expected to improve the variance (Chen et al., 2020; Huang et al., 2022). This surprising difference arises because those analyses focus on statistical errors arising from augmenting a size-$N$ real-life dataset to a size $Nk$ dataset, whereas our analysis pays attention to both statistical errors and computational errors in a setup that compares a size $N$ dataset versus a size $N/k \times k$ augmented dataset. Indeed, since the only computational saving of augmentation is the sampling cost $C_{\text{samp}} \ll C_{\text{optim}}$, every augmentation comes at the cost of one i.i.d. sample unused.

**Instability of DA is not specific to mean and variance**. While Proposition 4.1 only controls the mean and the variance, they do describe the distributions of $\delta\theta^{(\text{DA})}$ and $\delta\theta^{(\text{OG})}$ well, even *in the high-dimensional regime* where the number of parameters $q$ is large compared to the batch size $N$. This is due to recent results on high-dimensional Central Limit Theorem (CLT): In Theorem D.1 in the appendix, we adapt results from Chernozhukov et al. (2017) to show that $\delta\theta^{(\text{DA})}$ and $\delta\theta^{(\text{OG})}$ are approximately normal in an appropriate sense. This shows that the instability of DA parameter update is a general feature of the distribution, and not just specific to the mean and the variance.

**Applicability to multi-step updates**. Proposition 4.1 concerns one-step gradients, but the analysis applies to multi-step updates: VMC methods draws fresh samples at every step conditionally on the parameter $\theta$ from the previous step, so the same bounds hold with $\mathbb{E}[\bullet]$ and $\text{Var}[\bullet]$ replaced by the conditional counterparts $\mathbb{E}[\bullet | \theta]$ and $\text{Var}[\bullet | \theta]$.

One limitation of the above analysis is that it is restricted to first-order updates. Extending similar analyses to second-order updates is a known challenge in the literature: Those methods typically pre-multiply the empirical average $\frac{1}{N}\sum_{i \leq N/k}\sum_{j \leq k} F_{\mathbf{g}_{i,j}(\mathbf{X}_i);\psi_\theta}$ by the inverse of an empirical Fisher information matrix, which is also affected by augmentation, and there exist pathological examples where augmentation may increase or decrease the variance depending on problem-specific parameters (see e.g., ridge regression analysis in Huang et al. (2022)). Nevertheless, we confirm numerically in Fig. 3 that the gradient variance

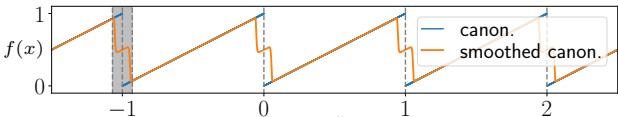

*Figure 4.* Canonicalization functions for 1d unit translations.

under KFAC, a second-order method used by DeepSolid, is also inflated under DA. Table 1 also verifies that the computational speedup from DA is negligible. Fig. 5 and Table 2 show that training with DA leads to a worse performance.

### 4.2. Group-averaging (GA)

Fix $\mathcal{G}$ with $|\mathcal{G}| = k$ and let $\psi_\theta^\mathcal{G}$ be the averaged network obtained from taking $\psi = \psi_\theta$ in (4). As with DA, since $k$-times more derivatives are computed, we need to use a batch size of $N/k$ to maintain the same computational cost. Draw $\mathbf{X}_1^\mathcal{G}, \ldots, \mathbf{X}_{N/k}^\mathcal{G} \overset{\text{i.i.d.}}{\sim} p^{(m)}_{\psi_\theta^\mathcal{G}}$. The GA update rule is

$$\theta \mapsto \theta - \delta\theta^{(\text{GA})}, \quad \delta\theta^{(\text{GA})} := \frac{1}{N/k}\sum_{i \leq N/k} F_{\mathbf{X}_i^\mathcal{G};\psi_\theta^\mathcal{G}}.$$

The mean and variance of $\delta\theta^{(\text{GA})}$ is straightforward:

**Lemma 4.2.** *Fix $\theta \in \mathbb{R}^p$. Then*
$$\mathbb{E}[\delta\theta^{(\text{GA})}] = \mathbb{E}[F_{\mathbf{X}_1^\mathcal{G};\psi_\theta^\mathcal{G}}], \quad \text{Var}[\delta\theta^{(\text{GA})}] = \frac{\text{Var}[F_{\mathbf{X}_1^\mathcal{G};\psi_\theta^\mathcal{G}}]}{N/k}.$$

*Remark* 4.3. As with DA, we can mean and variance as proxies to understand the distribution of $\delta\theta^{(\text{GA})}$ since a high-dimensional CLT does apply here; see Appendix D.

**GA may also destabilize gradients.** The decrease in sample size is again visible in the variance in Lemma 4.2, which suffers from a $\sqrt{k}$ blowup. Notice that the reference mean and variance are stated in terms of the gradient $F_{\mathbf{X}_1^\mathcal{G};\psi_\theta^\mathcal{G}}$, which depends on the GA wavefunction both through the samples $\mathbf{X}_1^\mathcal{G}$ and through the gradient evaluation at $\psi_\theta^\mathcal{G}$. Unlike the discussion in Proposition 4.1, we no longer expect that the mean and variance of $F_{\mathbf{X}_1^\mathcal{G};\psi_\theta^\mathcal{G}}$ are close to the unsymmetrized analogue $F_{\mathbf{X}_1;\psi_\theta}$, since it does not suffice for $\mathbf{X}_1^\mathcal{G}$ and $\mathbf{X}_1$ to have similar distributions. In general, $\text{Var}[\delta\theta^{(\text{GA})}]$ increases if and only if the ratio

$$\frac{\text{Var}[F_{\mathbf{X}_1^\mathcal{G};\psi_\theta^\mathcal{G}}]}{\text{Var}[F_{\mathbf{X}_1;\psi_\theta}]} > \frac{1}{k}.$$

Empirically, we see that a variance increase for $\delta\theta^{(\text{GA})}$ is visible for KFAC in Fig. 3 compared to $\delta\theta^{(\text{OG})}$ with similar computational costs (Table 1).

We also include two further comparisons:

**GA with subsampling (GAs).** One way to circumvent this computational hurdle is to average over a size-$k$ uniform subsample of $\mathcal{G}$ at every training step, and use the full $\mathcal{G}$ at inference time. We numerically investigate the effects of keeping $N$ constant and uniformly sample $k = 1$ element: While GAs further destabilizes the gradient (Fig. 3), Table 2

shows that its performance improves from OG. Yet, it falls short of the performance obtained by post hoc averaging directly on the original wavefunction.

**GA with same $N$.** Say the batch size $N$ is kept the same. While the per-step cost of a $k$-fold averaging increases by $\approx k$ times (Table 1), one may ask if the number of steps until convergence may be reduced under the symmetrized wavefunction, such that the overall training cost is constant. Fig. 3 shows that while such a reduction does appear, it is not enough to offset the increase in per-step cost.

### 4.3. Smoothed canonicalization (SC)

Another symmetrization method that gained traction in the theoretical ML community is canonicalization, defined as the projection to the fundamental region of a given group (Kaba et al., 2023; Dym et al., 2024). For 1d unit translations, an example of canonicalization is the map $x \mapsto x \bmod 1$, illustrated in blue in Fig. 4. Canonicalization for space groups $\mathbb{G}_{\mathrm{sp}}$ is possible but suffers from non-smoothness at the boundary, as visible at 0 and 1 in Fig. 4 and as we show in Appendix E. Jastrow factors (Whitehead et al., 2016) from VMC methods can be viewed as a way to smooth 1d canonicalization along each lattice vector, but the construction is specific to translations. Dym et al. (2024) proposes a smoothed canonicalization (SC) for large permutation and rotation groups by taking weighted averages at the boundary. In Appendix E, we adapt their idea to develop an SC for diagonal invariance under $\mathbb{G}_{\mathrm{sp}}$ that also respects the anti-symmetry constraint of (1). The orange curve in Fig. 4 illustrates our method for 1d translations. However, we demonstrate in Appendix E that SC via weighted averaging requires averaging over $n \times |\mathcal{G}_\epsilon|$ elements, where $\mathcal{G}_\epsilon$ is some carefully chosen subset of $\mathbb{G}_{\mathrm{sp}}$ and $n$ is the number of electrons; the additional cost of $n$ arises from an anti-symmetry requirement. Therefore, SC suffers from similar computational bottlenecks as DA and GA and typically to a worse extent. This renders SC unsuitable for training. Since SC for $\mathbb{G}_{\mathrm{diag}}$ may be of independent interest, we provide theoretical guarantees and a discussion of its weaknesses in Appendix E.

SC is an architecture-agnostic invariant map for $\mathbb{G}_{\mathrm{diag}}$, and its shortcomings demonstrate our discussion in Section 2 that obtaining invariant maps for such restricted symmetries can be difficult. An alternative approach (Han et al., 2019; Zepeda-Núñez et al., 2021; Gao & Günnemann, 2021; Gerard et al., 2022) is to introduce invariance implicitly, by building feature maps based on electron-atom distances. One must then ensure the network is invariant under permutations of atomic indices. Introducing this permutation invariance naively, by averaging, requires an expensive summation over a permutation group, and introduces additional complications with the boundary condition. The works

referenced above address this by building simpler early-layer features that reduce the amount of averaging required. However, this approach restricts the representation power of earlier layers, and is not directly compatible with the DeepSolid architecture. It remains unclear whether those architectures can be adapted to perform well in solid systems. We focus on DeepSolid for the purpose of our investigation.

## 5. Post hoc symmetrization

An alternative to in-training symmetrization is post hoc symmetrization: We may first train $\hat{\theta}$ with unsymmetrized updates (e.g. (6)), and seek to symmetrize $\psi_{\hat{\theta}}$ during inference. In contrast to Sec. 4, post hoc symmetrization no longer incurs the cost of $C_{\mathrm{grad}}$. While computing properties based on Monte Carlo estimates still incurs $C_{\mathrm{sample}}$, samples are typically obtained in large batches from long chains only once, before being used for multiple downstream computations. This allows us to perform a moderate amount of averaging without compromising on sample size. Meanwhile, a direct evaluation of the wavefunction, e.g. for producing the visualization in Fig. 1, also does not incur $C_{\mathrm{sample}}$.

A wavefunction $\psi_\theta$, as a physical object, is deterministic and should not involve exogenous randomness. As such, we do not consider DA for post-processing, and discuss only group averaging and canonicalization in this setup.

**Post hoc averaging (PA).** Since the cost of averaging still scales linearly with $|\mathbb{G}|$, taking an average over the entire $\mathbb{G}$ can still be prohibitive when $\mathbb{G}$ is large. Fortunately, Fact 2.1 ensures the validity of averaging over any finite subset $\mathcal{G} \subseteq \mathbb{G}$ of our choice. One may choose $\mathcal{G}$ to be a subgroup of interest, or a generating set of $\mathbb{G}$. Given a *trained* wavefunction $\psi_{\hat{\theta}}$, the corresponding PA wavefunction reads

$$\psi_{\hat{\theta}}^{(\mathrm{PA};\mathcal{G})}(\mathbf{x}) \; := \; \frac{1}{|\mathcal{G}|} \sum\nolimits_{g \in \mathcal{G}} \psi_{\hat{\theta}}(g(\mathbf{x})) \;.$$

Estimates of physical quantities are obtained by drawing $N$ samples from $p_{\psi_{\hat{\theta}}^{(\mathrm{PA};\mathcal{G})}}$, incurring a computational cost of $N |\mathcal{G}| C_{\mathrm{samp}}$ per MCMC step. For $\mathcal{G}$ of a moderate size, we can compare $N$ samples from $p_{\psi_{\hat{\theta}}^{(\mathrm{PA};\mathcal{G})}}$ directly with $N$ samples from $p_{\psi_{\hat{\theta}}}$, without incurring a larger statistical error. $\psi_{\hat{\theta}}^{(\mathrm{PA};\mathcal{G})}$ offers two clear advantages over $\psi_{\hat{\theta}}$:

(i) *More symmetry*. By construction, $\psi_{\hat{\theta}}^{(\mathrm{PA};\mathcal{G})}$ exhibits a higher degree of symmetry than $\psi_{\hat{\theta}}$, the extent of which depends on the subset $\mathcal{G}$ chosen and the degree of symmetry already present in $\psi_{\hat{\theta}}$. See Fig. 1 for the improved symmetry.

(ii) *Robustness to outliers*. The ground-truth wavefunctions are required to be non-smooth whenever an electron coincides with an atom or an electron. At those regions of $\mathbf{x}$, the Hamiltonian $H\psi_\theta(\mathbf{x})$ diverges, but for ground-truth wavefunctions as well as carefully constrained clas-

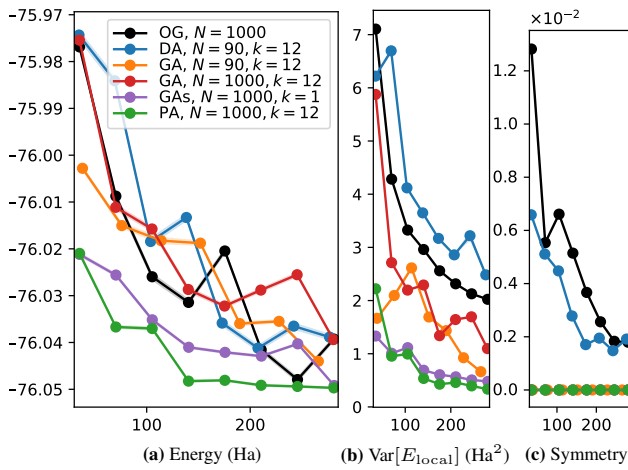

(a) Energy (Ha)  (b) Var$[E_{\mathrm{local}}]$ (Ha$^2$)  (c) Symmetry

*Figure 5.* Performance of wavefunctions against GPU hours, obtained with different symmetrization methods. Metrics are defined in Sec. 6 and experiment details in Sec. 7. (c) is computed via Var[PA/OG] in Sec. 6 but with $\psi_\theta^{(\mathrm{OG})}$ replaced by different $\psi_\theta$'s.

sical wavefunction ansatz, the local energy $E_{\mathrm{local};\psi_\theta}(\mathbf{x}) = H\psi_\theta(\mathbf{x})/\psi_\theta(\mathbf{x})$ in (5) stays finite due to the renormalization by $\psi_\theta(\mathbf{x})$. The same is not necessarily true for neural network solvers, which are much more flexible by design: The Laplacian term in $H\psi_\theta(\mathbf{x})$ may become large even when the magnitude of $\psi_\theta(\mathbf{x})$ stays put. This issue also manifests in the evaluation of other empirical quantities, e.g., the gradient terms computed in MCMC. The averaging in $\psi_{\hat{\theta}}^{(\mathrm{PA};\mathcal{G})}$ improves robustness in the sense that, if $\psi_{\hat{\theta}}(g(\mathbf{x}))$ takes a large numerical value for one particular $g \in \mathcal{G}$ and not for the rest, the numerical outlier is rescaled by a factor $\frac{1}{|\mathcal{G}|}$. The level of robustness also increases with the size of $\mathcal{G}$.

Fig. 5 and Table 2 show that $\psi_{\hat{\theta}}^{(\mathrm{PA};\mathcal{G})}$ outperforms in-training symmetrization with the same computational costs in all metrics considered. Compare, for example, PA with 40k steps of training with $N = 1000$ versus GA with 10k steps of training, $N = 1000$ and $k = 12$: PA achieves a lower energy with lower variance as well as perfect symmetry, with only $1/4$ of the training budget. Among methods with similar end-of-training energies, PA also attain a lower energy and variance with fewer training steps (Fig. 5). We also remark that GA and GAs by default implement PA at inference, so their only difference with PA is from training.

**Issues with post hoc canonicalization (PC).** One may also use smooth canonicalization post hoc to symmetrize a trained wavefunction $\psi_{\hat{\theta}}$. For completeness, we record the performance of PC in Table 2. The results are significantly worse than other methods, despite being applied to a well-trained wavefunction. The issue might arise from the fact that a weighted averaging near the boundary leads to a blowup in second derivatives, and we examine it in detail in Appendix E. This makes PC unsuitable specifically for our problem, since $E_{\mathrm{local}}$ involves the Hamiltonian.

## 6. Evaluation and visualization methods

VMC wavefunctions are typically assessed via

$$\mathbb{E}[E_{\mathrm{local};\psi_\theta}(\mathbf{X})] \text{ and } \mathrm{Var}[E_{\mathrm{local};\psi_\theta}(\mathbf{X})] , \mathbf{X} \sim p_{\psi_\theta} . \quad (7)$$

The energy is our optimization objective (5). The variance is another measure of fit for (1): It admits a lower bound 0 that is attained by any true solution $\psi_*$ to (1), since $E_{\mathrm{local};\psi_*}$ is everywhere constant. See Kent et al. (1999).

To show the amount of approximate symmetry already present in the OG wavefunction $\psi_\theta^{(\mathrm{OG})}$, we compare $\psi_\theta^{(\mathrm{OG})}$ against PA wavefunctions averaged over different $\mathcal{G}$'s. This is reported as Var[PA/OG] in Table 2, which stands for

$$\mathrm{Var}\left[\frac{1}{|\mathcal{G}|} \sum_{g \in \mathcal{G}} \psi_\theta^{(\mathrm{OG})}(g(\mathbf{X})) / \psi_\theta^{(\mathrm{OG})}(\mathbf{X})\right], \mathbf{X} \sim p_{\psi_\theta^{(\mathrm{OG})}} .$$

We also seek to visualize $\psi_\theta$ for its symmetry under $\mathbb{G}_{\mathrm{diag}}$. Visualizing diagonal symmetry can be challenging, as $\mathbb{G}_{\mathrm{diag}}$ acts on a high-dimensional space $\mathbb{R}^{3n}$. We propose a visualization method that exploits the fact that our $\mathbb{G}_{\mathrm{diag}}$ is completely described by a group $\mathbb{G}$ of isometries in $\mathbb{R}^3$:

(i) Let $\tilde{\mathbb{G}}$ be a group acting on $\mathbb{R}^3$ defined as $\tilde{\mathbb{G}} := \left\{ A \in \mathbb{R}^{3 \times 3} \,\middle|\, A(\bullet) + b \in \mathbb{G} \text{ for some } b \in \mathbb{R}^3 \right\}$ ;

(ii) Fix $\tilde{\mathbf{x}}_{\mathrm{symm}} \in \mathbb{R}^{3n}$, a configuration of $n$ electrons such that for every $g \in \tilde{\mathbb{G}}$, $g(\tilde{\mathbf{x}}_{\mathrm{symm}}) = \tilde{\mathbf{x}}_{\mathrm{symm}}$ ;

(iii) Given a function $f : \mathbb{R}^{3n} \to \mathbb{R}$ to visualize, we plot the function $\tilde{f}(t) := f(\tilde{\mathbf{x}}_{\mathrm{symm}} + t)$ with $t \in \mathbb{R}^3$, i.e. all electrons are translated by $t$ simultaneously.

The next result confirms the validity of this method:

**Lemma 6.1.** *Let $f : (\mathbb{R}^3)^n \to \mathbb{R}$ be a function invariant under permutations of its $n$ arguments. Then for any $g \in \mathbb{G}$ and $t \in \mathbb{R}^3$, $\tilde{f}(g(t)) - \tilde{f}(t) = f(g(\mathbf{x} + t)) - f(\mathbf{x} + t)$ .*

The permutation invariance assumption holds for $|\psi_\theta|$ and $E_{\mathrm{local};\psi_\theta}$ since $\psi_\theta$ is anti-symmetric. Fig. 1(a), (b) and (f) are plotted with this method, with $f(\mathbf{x}) = \log|\psi_\theta(\mathbf{x})|^2$, $\tilde{\mathbf{x}}_{\mathrm{symm}}$ given in Fig. 1(e) and $\mathbb{G} = \tilde{\mathbb{G}} = \mathrm{P3m1}$, which illustrates the *partial* symmetries of the P6mm group of graphene. To see the $\mathbb{G} = \mathrm{P6mm}$ symmetry, a different $\tilde{\mathbf{x}}_{\mathrm{symm}}$ is required since $\tilde{\mathbb{G}} \neq \mathbb{G}$ in this case; see Appendix C.

## 7. Experimental details

All code and data are available at github.com/ PrincetonLIPS/invariant-DeepSolid. Experiments are performed with DeepSolid (Li et al., 2022) on crystalline solids. Each network is evaluated by sampling from MCMC chains with 30k length, and the model from the last training step is used unless otherwise specified. Supercell size is included in the first column of Table 2. Appendix B includes further specifications, experiments and a remark about why energy improvements in the decimals are considered substantial. A few remarks about each system:

| System | $\mathcal{G}$ | Method | $N$ | $k$ | Steps | GPU hours* | Energy (Ha) | $\mathrm{Var}[E_{\mathrm{local}}]$ (Ha²) | $\mathrm{Var}[\mathrm{PA/OG}]$ |
|---|---|---|---|---|---|---|---|---|---|
| Graphene $1 \times 1$ | P6mm | OG | 1000 | - | 80,000 | 281 | $-76.039(6)$ | 2.02(3) | |
| | | DA | 90 | 12 | 80,000 | 277 | $-76.039(3)$ | 2.48(5) | |
| | | GA | 90 | 12 | 80,000 | 304 | $\mathbf{-76.049(3)}$ | 0.58(2) | |
| | | GA | 1000 | 12 | 10,000 | 351 | $-76.034(5)$ | 1.2(2) | |
| | | GAs | 1000 | 1 | 80,000 | 281 | $\mathbf{-76.049(3)}$ | 0.48(2) | |
| | | PA | 1000 | 12 | 80,000 | 281 | $\mathbf{-76.050(3)}$ | $\mathbf{0.33(1)}$ | 0.00180(4) |
| | | PC | 1000 | 12 | 80,000 | 281 | $-70.1(9)$ | $8(1) \times 10^2$ | |
| Lithium Hydride (LiH) $2 \times 2 \times 2$ | - | OG | 4000 | - | 30,000 | 571 | $-8.138(2)$ | 0.06(1) | |
| | P$\bar{1}$ | PA | | 2 | | | $-8.144(1)$ | 0.0344(9) | 0.0183(4) |
| | P2/m | PA | | 4 | | | $-8.148(1)$ | 0.0197(6) | 0.0249(5) |
| | F222 | PA | | 16 | | | $\mathbf{-8.1495(9)}$ | 0.0162(7) | 0.0265(5) |
| | Pm$\bar{3}$m | PA | | 48 | | | $\mathbf{-8.1502(7)}$ | $\mathbf{0.0122(7)}$ | 0.0289(8) |
| | Fm$\bar{3}$m | PA | | 192 | | | $\mathbf{-8.1507(8)}$ | $\mathbf{0.0118(7)}$ | 0.0289(8) |
| Metallic Lithium (bcc-Li) $2 \times 2 \times 2$ | - | OG | 3000 | - | 20,000 | 462 | $-15.011(1)$ | 0.059(2) | |
| | P4/mmm | PA | | 16 | | | $\mathbf{-15.021(2)}$ | $\mathbf{0.033(2)}$ | 0.092(5) |
| | Fmmm | PA | | 32 | | | $\mathbf{-15.020(1)}$ | 0.036(2) | 0.0101(4) |
| | Im$\bar{3}$m | PA | | 96 | | | $\mathbf{-15.022(3)}$ | $\mathbf{0.031(3)}$ | 0.0139(6) |

*Table 2.* Performance of symmetrization methods with similar computational budgets. Energy and variance are both reported at the per unit cell level. *See Appendix B.3 for specifications of the GPUs used for training.

**Graphene.** This is the setup considered in Fig. 1,3,5 and Table 1. PA outperforms other methods both in terms of the metrics in Table 2 and speed of convergence in Fig. 5.

**LiH.** We use nested subgroups of Fm$\bar{3}$m and observe that the performance improves with $k$. For comparison, Li et al. (2022) report the energy $-8.15096(1)$ for DeepSolid trained with 3e5 steps and batch 4096. PA attains comparable performance in similar systems[1] with 3e4 steps and batch 4000.

**bcc-Li.** This is a known difficult case for ab initio methods including DeepSolid (Yao et al., 1996; Li et al., 2022) and PA again helps. Fmmm and P4/mmm are subgroups of Im$\bar{3}$m containing different symmetries, and each offers similar improvements. For both LiH and bcc-Li, we also observe a saturation effect: The improvement saturates once sufficiently many symmetries are incorporated. We do not know whether an optimal choice of subgroup exists that balances performance and computational cost.

## 8. Discussion

For ML problems that exhibit geometric structure, conventional wisdom holds that incorporating the correct symmetries should improve performance. This is called into question by the recent success of AlphaFold3 (Abramson et al., 2024), where substantial accuracy improvement is obtained without incorporating invariance into its architecture, as well as by other works on atomic potential and ML theory referenced in Section 1. Our work investigates such tradeoffs in the context of wavefunctions, and specifically of VMC. To this end, we compare different model-agnostic symmetrizations. To ensure comparability, we do so on a single architecture, and choose hyperparameters as in the original DeepSolid work (Li et al., 2022). We do not claim to have exhausted all possible ways in which symmetries may affect VMC.

Regarding the implications of our findings for other settings, we note a number of points:

**Computational-statistical tradeoffs for DA and GA.** We expect the in-training tradeoffs observed in Section 4 to be applicable to other ML setups where sampling is performed in between gradient updates. A non-physics example is the contrastive divergence algorithm for training energy-based models (Hinton, 2002; Du et al., 2020).

**Computational cost of SC.** The computational cost discussed in Section 4.3 is specific to the canonicalization we have adopted from Dym et al. (2024), which adopts an "average-near-the-boundary" approach. It is an open question whether more efficient canonicalizations exist for $\mathbb{G}_{\mathrm{sp}}$.

**Applicability beyond DeepSolid and VMC.** We conjecture that both our negative in-training findings and positive post-training findings are applicable beyond DeepSolid and VMC, especially in cases where difficult cases of restricted symmetries arise. As discussed in Section 2, if the system of interest possesses simpler symmetries such as translations or $E(3)$, there often exist more efficient symmetrizations than averaging or augmentation. These may avoid the in-training statistical-computational tradeoffs we observe.

**Theoretical analyses.** While our experiments focus on DeepSolid, our theoretical analyses describe more generally how DA, GA and SC interact with VMC.

---

[1]The energy by Li et al. (2022) is for lattice vector $4.0$ Å. We followed the Materials Project (Jain et al., 2013) to use $4.02$ Å.

## Acknowledgement

This work was partially supported by NSF OAC 2118201. KHH and PO are supported by the Gatsby Charitable Foundation (GAT3850). NZ acknowledges support from the Princeton AI$^2$ initiative. This work used Princeton ionic cluster and Delta GPU at the National Center for Supercomputing Applications through allocation MAT220011 from the Advanced Cyberinfrastructure Coordination Ecosystem: Services & Support (ACCESS) program, which is supported by National Science Foundation grants #2138259, #2138286, #2138307, #2137603, and #2138296. We also thank an anonymous reviewer for pointing out regimes where per-step sampling cost could be larger than per-step optimisation cost.

## Impact Statement

This paper presents work whose goal is to advance the field of Machine Learning. There are many potential societal consequences of our work, none which we feel must be specifically highlighted here.

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

The appendix is organized as follows:

- Appendix A collects the notation used throughout the paper.

- Appendix B includes additional figures and details on the experiments.

- Appendix C proves the validity of the diagonal visualization method (Lemma 6.1) and includes additional details on how to visualize the full symmetry beyond Fig. 1.

- Appendix D states the high-dimensional central limit theorem discussed in Section 4.

- Appendix E develops and discusses the smoothed canonicalization method discussed in Sections 4.3 and 5 for diagonal invariance under a space group.

- Appendices F to H prove all mathematical results developed in this paper.

- Appendix I includes an additional discussion on the case where per-iter sampling cost is greater than per-iter gradient cost.

## A. Notation

- $n$: number of electrons

- $N$: number of Markov chains / number of Markov chain samples

- $m$: length of Markov chain run

- $k$: number of symmetry operations

- $x$: $\mathbb{R}^3$-valued position of a single electron. $\tilde{x} = (x, \sigma) \in \mathbb{R}^3 \times \{\uparrow, \downarrow\}$ additionally specifies the spin $\sigma$ of the electron.

- $\mathbf{x}$: $\mathbb{R}^{3n}$-valued positions of $n$ electrons, also called an electron configuration. The diagonally transformed configuration is denoted as $g(\mathbf{x}) = (g(x_1), \ldots, g(x_n))$. Similarly, $\tilde{\mathbf{x}} \in (\mathbb{R}^3 \times \{\uparrow, \downarrow\})^n$ is the $n$-electron analogue of $\tilde{x}$.

- $\mathbb{G}$: Space group acting on $\mathbb{R}^3$

- $\mathbb{G}^*$: Point group subgroup of $\mathbb{G}$, consisting of elements of the form $x \mapsto A(x)$ where $A$ is an orthogonal $3 \times 3$ matrix

- $\mathbb{G}_{\text{diag}}$: Diagonal group induced by the space group, acting on an electron configuration in $\mathbb{R}^{3n}$

- $\mathcal{G}$: subset (not necessarily a subgroup) of elements of $\mathbb{G}_{\text{diag}}$

- $P_n$: permutation group acting on an electron configuration in $\mathbb{R}^{3n}$

- $\mathbb{T}_{\text{sup}}$: translation group acting on $\mathbb{R}^3$ that represents the supercell assumption

- $\otimes$: the tensor product

- $\| \bullet \|$: Euclidean norm

- $(\bullet)_l$: $l$-th coordinate of a vector

- absolute constant: a number that does not depend on any other variables, and in particular not on $n$, $N$, $k$ or $M$

- $\overset{d}{=}$: equality in distribution

## B. Experimental details and additional results

### B.1. DeepSolid architecture

All our experiments use DeepSolid (Li et al., 2022) as $\psi_\theta$, the baseline unsymmetrized network. DeepSolid adapts FermiNet, a molecular neural network ansatz (Pfau et al., 2020), to infinite periodic solids. We briefly review their architectures.

Architecture-wise, FermiNet takes as inputs $n$ electron positions and split them into two groups according to spins. For some distance function $d$, the electrons are encoded through feature maps of the form $d(x_j - x_k)$ and $d(x_j - r_K)$, i.e. electron-electron interactions and electron-nucleus interactions, which are passed into a fully-connected network with $\tanh$ activations.

The outputs are orbital information at the single-electron level that depends on electron-electron interactions, which are combined by a weighted sum of Slater determinants to form antisymmetric many-electron wavefunctions. For details, see Fig. 1 and Algorithm 1 of (Pfau et al., 2020).

DeepSolid makes several adaptations of FermiNet that are crucial for modelling periodic solids. Of those, the most relevant one to our discussion are their choice of periodic features $d(x_j - x_k)$ and $d(x_j - x_K)$, stated in (2) of (Pfau et al., 2020). Their periodic features introduce separate invariance with respect to supercell translations, such that the $n$ electrons are effectively restricted within the supercell. The construction is based on Jastrow correlation factors (Whitehead et al., 2016) and can be viewed as a version of smoothed canonicalization for 1d translations (Section 4.3).

## B.2. Physical systems

| Physical systems | Graphene $1 \times 1$ | LiH $2 \times 2 \times 2$ | bcc-Li $2 \times 2 \times 2$ |
|---|---|---|---|
| Number of electrons | 12 | 32 | 48 |
| Point group symmetry of the system | P 6mm | Fm$\bar{3}$m | Im$\bar{3}$m |
| Number of point group elements | 12 | 192 | 96 |
| Lattice vector length (conventional) | 2.4612 Å | 4.02 Å | 3.4268 Å |

Table 3. List of physical systems considered in the experiments, where the number $a \times b$ indicates the size of the supercell for a 2d system and $a \times b \times c$ is that for a 3d system. The primitive cell (corresponding to $1 \times 1 \times 1$ supercell) of graphene is visualized in Fig. 1(d), and those of LiH and bcc-Li are in Fig. 6(a) and Fig. 6(e).

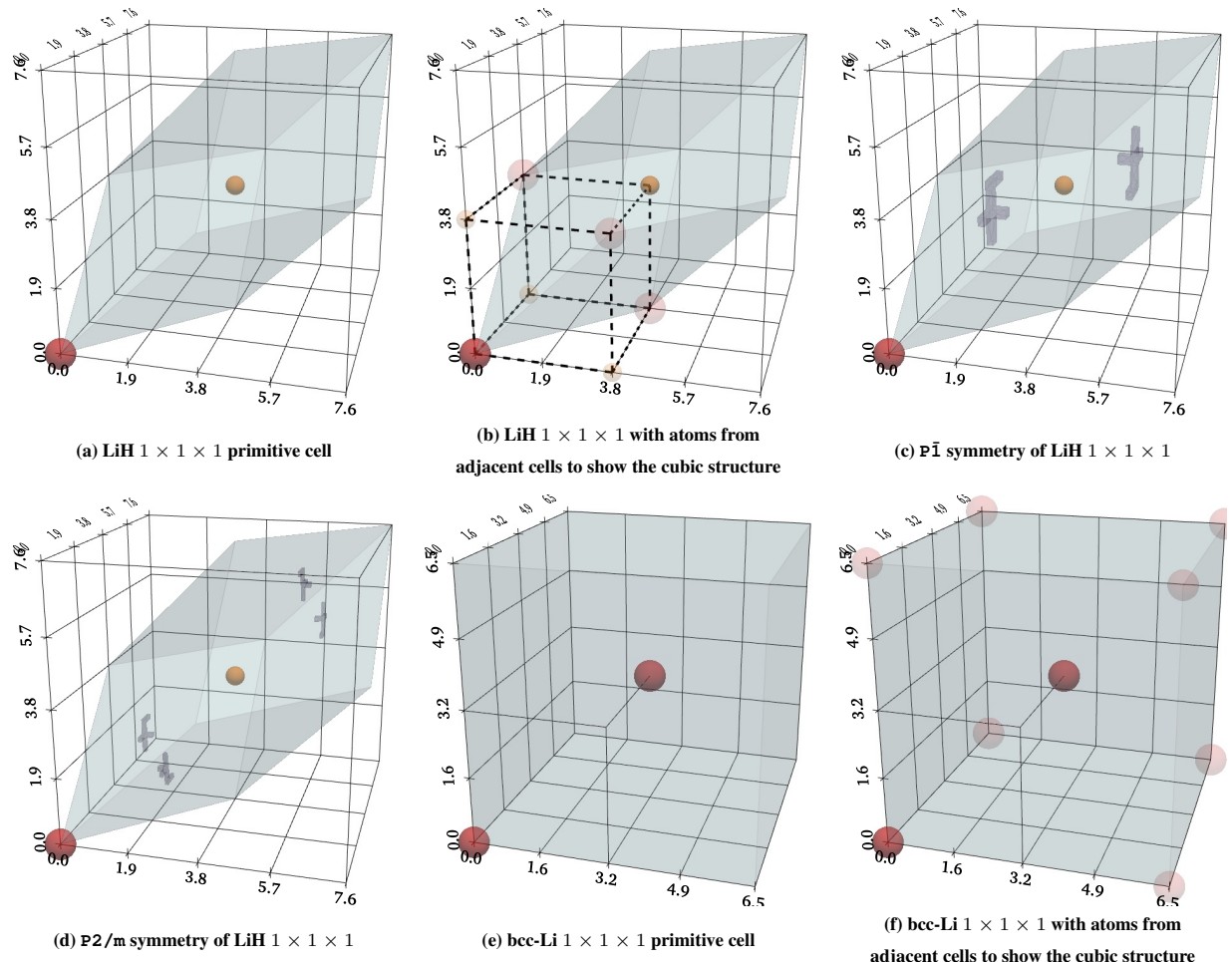

(a) LiH $1 \times 1 \times 1$ primitive cell

(b) LiH $1 \times 1 \times 1$ with atoms from adjacent cells to show the cubic structure

(c) P$\bar{1}$ symmetry of LiH $1 \times 1 \times 1$

(d) P2/m symmetry of LiH $1 \times 1 \times 1$

(e) bcc-Li $1 \times 1 \times 1$ primitive cell

(f) bcc-Li $1 \times 1 \times 1$ with atoms from adjacent cells to show the cubic structure

Figure 6. Visualization of the primitive cells of LiH and bcc-Li.

The experimental benchmark used for comparing all symmetrization strategies is graphene with an $1 \times 1$ supercell for

computational convenience. Effects of post-hoc averaging over different choices of $\mathcal{G}$ are illustrated through LiH and bcc-Li, which possess substantially more symmetries. Two types of simple symmetries in LiH, `P1̄` and `P2/m`, are illustrated in Fig. 6(c) and (d); we refer readers to (Brock et al., 2016) for an exhaustive visualization of all 2d and 3d space groups.

### B.3. Parameter settings

The symmetrization strategies we consider, i.e. DA, GA, SC, PA and PC, are all implemented as wrapper functions around the original DeepSolid model. Architecturally this can be viewed as inserting one layer each before and after the DeepSolid processing pipeline. Notably, this allows us to leave most of the training parameters from DeepSolid unchanged and retain the performance from the original DeepSolid.

To ensure a fair comparison, we keep the default network and training parameter settings from DeepSolid across all our experiments, except that we vary the training batch size according to $k$, the number of symmetry operations used. Graphene training uses NVIDIA GeForce RTX 2080 Ti (12GB) and LiH and bcc-Li use NVIDIA A100 SXM4 (40GB).

At the inference stage, we collect samples from independent MCMC chains with length $30,000$. The numbers of samples collected are respectively $50,000$ for graphene and $20,000$ for LiH and bcc-Li. The choice of smoothing in post-hoc canonicalization is $s_\infty$ defined in Appendix E.1; see Appendix E for details.

### B.4. Additional results

**A remark about the scale of energy improvements.** Notice that the energy improvements reported in Table 2 are in the third decimals in Hartree. Improvements at this scale are crucial in physics and chemistry. In the physical systems we consider, core electron binding energies are on the order of keV (1 keV $\approx$ 37 Hartree), while valence electron binding energies are in the range of $1 - 10$ eV ($\approx 0.037$ Hartree). To obtain wavefunctions with accurate ground state energy, it is therefore crucial to obtain energy improvements on the order of $10^{-3}$ Hartree or smaller. In fact, the term "chemical accuracy" – used in computational chemistry to describe the level of precision needed for calculated energies to provide meaningful predictions of chemical phenomena, is given by 1 kcal / mol ($\approx 0.00159$ Hartree). For more references, see Williams et al. (2020); Perdew et al. (1996).

**Varying number of subsamples in GAs.** In Table 2 and Fig. 5, group-averaging with subsampling (GAs) achieves the closest performance to post hoc averaging on graphene $1 \times 1$. In Fig. 7 below, we verify that the performance does not improve if one varies $k$, the number of subsampled group elements, and that a larger $k$ also leads to gradient destabilization.

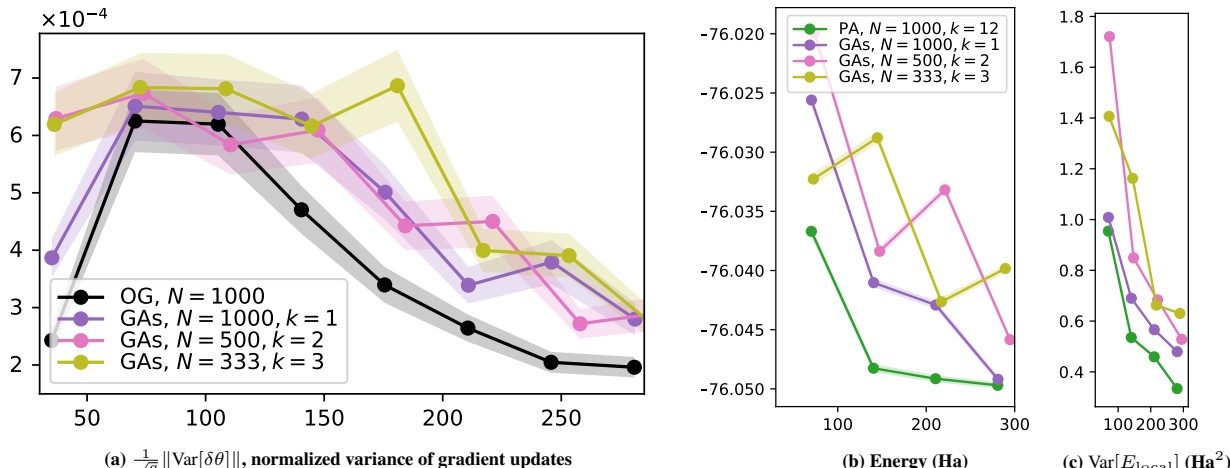

(a) $\frac{1}{\sqrt{q}} \|\mathrm{Var}[\delta\theta]\|$, **normalized variance of gradient updates**

(b) **Energy (Ha)**

(c) $\mathrm{Var}[E_{\mathrm{local}}]$ **(Ha$^2$)**

*Figure 7.* Performance of GAs plotted against GPU hours and across different subsample size $k$.

**Averaging over diagonal translations.** The results in Table 2 focus on averaging over point groups, and one may ask whether averaging additionally over translations helps. We perform a preliminary investigation in LiH and bcc-Li, and find that incorporating translations does not lead to significant performance improvement. The results are indicated in Table 4 by $\times\mathbb{T}$, and we include the corresponding results for point groups without translations for comparison. Note that as averaging over translations incurs $8\times$ of the computational cost, we have computed the statistics only on $10,000$ samples.

**Averaging over subsets of group elements.** Fact 2.1 ensures the validity of averaging over any *subset* of group elements

| System | $\mathcal{G}$ | Method | $N$ | $k$ | Steps | GPU hours | Energy (Ha) | Var$[E_{\text{local}}]$ (Ha$^2$) |
|---|---|---|---|---|---|---|---|---|
| LiH $2 \times 2 \times 2$ | - | OG | 4000 | - | 30,000 | 571 | $-8.138(2)$ | $0.06(1)$ |
| | $P m \bar{3} m$ | PA | | 48 | | | $\mathbf{-8.1502(7)}$ | $\mathbf{0.0122(7)}$ |
| | $P m \bar{3} m \times \mathbb{T}$ | PA | | 384 | | | $-8.1488(7)$ | $0.012(1)$ |
| bcc-Li $2 \times 2 \times 2$ | - | OG | 3000 | - | 20,000 | 462 | $-15.011(1)$ | $0.059(2)$ |
| | $P 4/mmm$ | PA | | 16 | | | $\mathbf{-15.021(2)}$ | $\mathbf{0.033(2)}$ |
| | $P 4/mmm \times \mathbb{T}$ | PA | | 128 | | | $-15.017(6)$ | $0.05(2)$ |

*Table 4.* Performance of post hoc averaging with translations.

and not just subgroups. The only caveat is that for a general finite subset $\mathcal{G}$, the average $\frac{1}{|\mathcal{G}|} \sum_{g \in \mathbb{G}} \psi(g(x))$ is no longer guaranteed to be invariant under $\mathcal{G}$. Nevertheless, one may still ask if averaging over subsets of a group $\mathbb{G}$ that are "sufficiently representative", e.g. the generators of the group $\mathbb{G}$, can be helpful. We perform a preliminary investigation on graphene and LiH for PA with $\text{Gen}(\mathbb{G})$, a fixed set of generators of $\mathbb{G}$ plus the identity element. The statistics are computed on $40,000$ samples and reported in Table 5. The results are inconclusive: We find that for graphene, PA with $\text{Gen}(\mathbb{G})$ improves energy but significantly inflates variance, whereas for LiH, PA with $\text{Gen}(\mathbb{G})$ has worse energy and variance. As a sanity check, we also compute $\text{Var}[\text{PA}^{\mathbb{G}}/\psi_{\theta}]$ for each wavefunction to verify that the PA with $\text{Gen}(\mathbb{G})$ is closer to PA computed on $\mathbb{G}$ compared to the original wavefunction.

| System | $\mathcal{G}$ | Method | $N$ | $k$ | Steps | GPU hours | Energy (Ha) | Var$[E_{\text{local}}]$ (Ha$^2$) | Var$[\text{PA}/\psi_{\theta}]$ |
|---|---|---|---|---|---|---|---|---|---|
| Graphene $1 \times 1$ | - | OG | 1000 | - | 80,000 | 281 | $-76.039(6)$ | $2.02(3)$ | $1.80(4) \times 10^{-3}$ |
| | $P 6mm$ | PA | | 12 | | | $-76.050(3)$ | $\mathbf{0.33(1)}$ | $\mathbf{0.0}$ |
| | $\text{Gen}(P 6mm)$ | PA | | 4 | | | $\mathbf{-76.064(5)}$ | $1.04(2)$ | $3.3(1) \times 10^{-4}$ |
| LiH $2 \times 2 \times 2$ | - | OG | 4000 | - | 30,000 | 571 | $-8.138(2)$ | $0.06(1)$ | $2.65(5) \times 10^{-2}$ |
| | $F 222$ | PA | | 16 | | | $\mathbf{-8.1495(9)}$ | $\mathbf{0.0162(7)}$ | $\mathbf{0.0}$ |
| | $\text{Gen}(F 222)$ | PA | | 5 | | | $-8.1456(7)$ | $0.0235(5)$ | $6.6(1) \times 10^{-3}$ |

*Table 5.* Performance of post hoc averaging with subsets of group elements.

## C. Additional details on Lemma 6.1 and visualizing diagonal invariance

We first state the proof of Lemma 6.1, which is illustrative for understanding the necessity of using a modified group $\tilde{\mathbb{G}} = \{A \in \mathbb{R}^{3 \times 3} \mid A(\bullet) + b \in \mathbb{G} \text{ for some } b \in \mathbb{R}^3\}$.

*Proof of Lemma 6.1.* By the definition of $\tilde{f}$,

$$\tilde{f}(g(t)) = f(\tilde{\mathbf{x}}_{\text{symm}} + A(t) + b) \overset{(a)}{=} f(A(\tilde{\mathbf{x}}_{\text{symm}}) + A(t) + b) \overset{(b)}{=} f(A(\tilde{\mathbf{x}}_{\text{symm}} + t) + b) = f(g(\mathbf{x} + t)).$$

In $(a)$, we have used that $\tilde{\mathbf{x}}_{\text{symm}}$ is invariant under $A \in \tilde{\mathbb{G}}$. Using the definition of $\tilde{f}$ again to note that $\tilde{f}(t) = f(\mathbf{x} + t)$ finishes the proof. $\square$

As mentioned in Section 6, $\tilde{\mathbb{G}}$ and the point group $\mathbb{G}_*$ of $\mathbb{G}$ are in general two different groups. When $\mathbb{G}_*$ is the $P 6mm$ point group, both $\mathbb{G}_*$ and $\tilde{\mathbb{G}}$ consist of 12 elements. However, $\mathbb{G}_*$ is generated by $P 3m1$ and the reflection $R$ indicated in Fig. 8, whereas $\tilde{\mathbb{G}}$ is generated by $P 3m1$ and the reflection $\tilde{R}$ indicated in Fig. 8. Applying our method to $\tilde{\mathbb{G}}$ allows for visualizing the full $P 6mm$ symmetry of the wavefunction (Fig. 9).

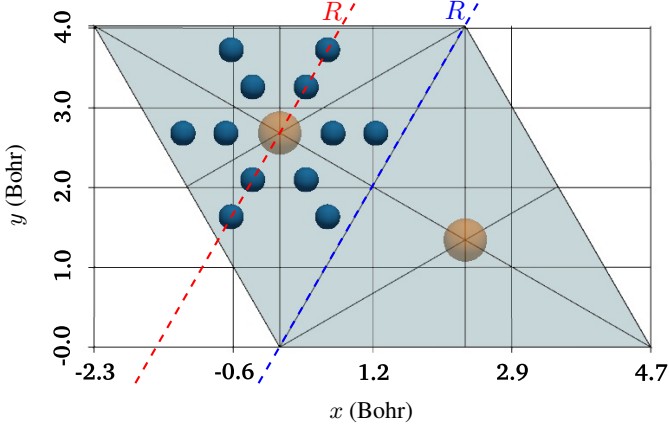

*Figure 8.* A $\tilde{\mathbb{G}}$-symmetric configuration of 12 electrons, where $\tilde{\mathbb{G}}$ is obtained from $P 6mm$

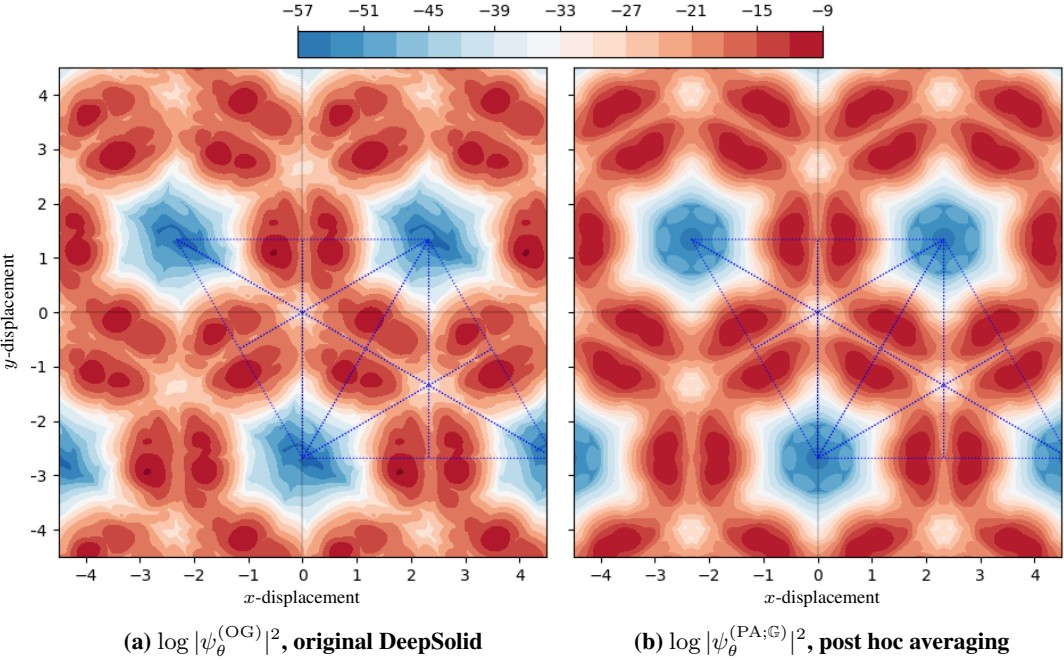

**(a)** $\log |\psi_\theta^{(\mathrm{OG})}|^2$, **original DeepSolid**  **(b)** $\log |\psi_\theta^{(\mathrm{PA};\mathbb{G})}|^2$, **post hoc averaging**

*Figure 9.* Visualization of the full $\mathbb{P}6mm$-diagonal invariance of $\psi_\theta^{(\mathrm{OG})}$ versus $\psi_\theta^{(\mathrm{PA};\mathbb{G})}$. Same setup and wavefunctions as Fig. 1, except that the configuration to be translated, $\mathbf{x}_{\mathrm{symm}}$, is given by Fig. 8.

## D. CLT results for DA and GA

We present results that characterize the distributions of the parameter update under data augmentation and group-averaging. First focus on the parameter update under DA:

$$\delta\theta^{(\mathrm{DA})} = \frac{1}{N} \sum_{i \leq N/k} \sum_{j \leq k} F_{\mathbf{g}_{i,j}(\mathbf{X}_i);\psi_\theta} \ ,$$

where $\mathbf{X}_1, \ldots, \mathbf{X}_{N/k} \overset{\mathrm{i.i.d.}}{\sim} p_{\psi_\theta}^{(m)}$ and $\mathbf{g}_{1,1}, \ldots, \mathbf{g}_{N/k,k}$ are i.i.d. samples from some distribution on $\mathbb{G}$.

Notice that due to augmentations, $\delta\theta^{(\mathrm{DA})}$ involves a correlated sum. Nevertheless, for each $l \leq p$, we can re-express the $l$-th coordinate of the DA parameter update as

$$\delta\theta_l^{(\mathrm{DA})} = \frac{1}{N/k} \sum_{i \leq N/k} F_{il}^{(\mathrm{DA})} \qquad \text{where } F_{il}^{(\mathrm{DA})} := \frac{1}{k} \sum_{j=1}^{k} \left( F_{\mathbf{g}_{1,j}(\mathbf{X}_1);\psi_\theta} \right)_l \ ,$$

an empirical average of i.i.d. univariate random variables across $1 \leq i \leq N/k$, which allows for the application of a CLT. A coordinate-wise CLT, i.e. the normal approximation of $\delta\theta_l^{(\mathrm{DA})}$ for any fixed $l \leq p$, is straightforward from classical CLT results. Since we are concerned about stability of the gradient estimate, it is more crucial to study the behavior of the coordinate of maximum deviation, i.e.

$$\max_{l \leq p} \left| \delta\theta_l^{(\mathrm{DA})} - \mathbb{E}[\delta\theta_{1l}^{(\mathrm{DA})}] \right| \ ,$$

and verify that its behavior is completely described by the mean and the variance. The next result complements Proposition 4.1 by providing CLT results for both individual coordinates and the coordinates of maximum deviation *in the high-dimensional regime*, where parameter dimension $p$ — the number of weights in our neural network — is allowed to be much larger than the batch size $N/k$ — the number of Markov chains per training step. For convenience, we denote the standard deviation of $F_{il}^{(\mathrm{DA})}$ as

$$\sigma_l^{(\mathrm{DA})} := \sqrt{\mathrm{Var}[F_{il}^{(\mathrm{DA})}]} = \sqrt{\frac{\mathrm{Var}[(F_{\mathbf{g}_{1,1}(\mathbf{X}_1);\psi_\theta})_l]}{k} + \frac{(k-1)\mathrm{Cov}[(F_{\mathbf{g}_{1,1}(\mathbf{X}_1);\psi_\theta})_l, (F_{\mathbf{g}_{1,2}(\mathbf{X}_1);\psi_\theta})_l]}{k}} \ .$$

**Theorem D.1.** *Fix $\theta \in \mathbb{R}^q$. Let $\mathbf{Z} \sim \mathcal{N}(0, I_p)$ be a standard Gaussian vector and $Z_l$ be its $l$-th coordinate. Then there exists some absolute constant $C_1$ such that for every $l \leq p$,*

$$\sup_{t \in \mathbb{R}} \left| \mathbb{P}\big( \delta\theta_l^{(\mathrm{DA})} \leq t \big) - \mathbb{P}\Big( \mathbb{E}\big[\delta\theta_l^{(\mathrm{DA})}\big] + (\mathrm{Var}[\delta\theta_l^{(\mathrm{DA})}])^{1/2} Z_l \leq t \Big) \right| \leq \frac{C_1 \, \mathbb{E}|F_{1l}^{(\mathrm{DA})}|^3}{\sqrt{N/k} \, (\sigma_l^{(\mathrm{DA})})^3} .$$

*Moreover, assume a mild tail condition that for every $l' \leq p$ with $\sigma_{l'}^{(\mathrm{DA})} > 0$, we have*

$$\mathbb{E}\left[ \exp\left( \frac{(\sigma_{l'}^{(\mathrm{DA})})^{-1} |F_{il}^{(\mathrm{DA})}|}{\tilde{F}^{(\mathrm{DA})}} \right) \right] \leq 2 \quad where \quad \tilde{F}^{(\mathrm{DA})} := \max_{\substack{l \leq p \\ \text{with } \sigma_l^{(\mathrm{DA})} > 0}} \max_{q \in \{3,4\}} \left\{ \left( \mathbb{E}\big[ (\sigma_l^{(\mathrm{DA})})^{-q} \, |F_{il}^{(\mathrm{DA})}|^q \big] \right)^{1/q} \right\} .$$

*Then the coordinate of maximum deviation of $\theta_1^{(\mathrm{DA})}$ also satisfies a CLT: There is some absolute constant $C_2 > 0$ such that*

$$\sup_{t \in \mathbb{R}} \left| \mathbb{P}\Big( \max_{l \leq p} |\delta\theta_l^{(\mathrm{DA})} - \mathbb{E}[\delta\theta_l^{(\mathrm{DA})}]| \leq t \Big) - \mathbb{P}\Big( \max_{l \leq p} \left| (\mathrm{Var}[\delta\theta_l^{(\mathrm{DA})}])^{1/2} Z_l \right| \leq t \Big) \right| \leq C_2 \left( \frac{(\tilde{F}^{(\mathrm{DA})})^2 \, (\log(pN/k))^7}{N/k} \right)^{1/6} .$$

The bounds in Theorem D.1 each control a difference in distribution function between $\delta\theta^{(\mathrm{DA})}$ and a normal distribution, measured through either an arbitrarily fixed coordinate or the coordinate of maximum deviation from its mean. In particular, they say that the distribution of $\delta\theta^{(\mathrm{DA})}$ is approximately normal and therefore completely characterized by the mean and the variance studied in Proposition 4.1. To interpret them in details:

- The coordinate-wise bound follows from the classical Berry-Esséen Theorem (see e.g. Theorem 3.7 of (Chen et al., 2011)), and the normal approximation error does not involve the dimension $p$.

- For the coordinate of maximum deviation, the normal approximation error is small so long as $p$ is much smaller than a constant multiple of $\exp((N/k)^{1/7})$, which is in particular true even if the parameter dimension $p$ is larger than the batch size $N/k$. The moment assumption amounts to a light-tailed condition on the distribution of the updates, and they can be relaxed at the cost of more complicated bounds — see Chernozhukov et al. (2017).

We also remark that Theorem D.1 is a special case of the universality result of Huang et al. (2022) for data augmentation, but with a sharper bound and in Kolmogorov distance.

Analogous CLTs hold for both the unaugmented update and the group-averaging update. Recall that these updates are

$$\delta\theta^{(\mathrm{OG})} := \frac{1}{N} \sum_{i \leq N} F_{\mathbf{x}_i; \psi_\theta} \qquad \text{and} \qquad \delta\theta^{(\mathrm{GA})} := \frac{1}{N/k} \sum_{i \leq N/k} F_{\mathbf{X}_i^{\mathcal{G}}; \psi_\theta} ,$$

where $\mathbf{X}_1^{\mathcal{G}}, \ldots, \mathbf{X}_{N/k}^{\mathcal{G}} \overset{\text{i.i.d.}}{\sim} p_{\psi_\theta^{\mathcal{G}}}^{(m)}$. In particular, $\delta\theta^{(\mathrm{OG})}$ is the same as $\delta\theta^{(\mathrm{DA})}$ with $k = 1$ and $\mathbf{g}_{1,1}$ set to the identity transformation, and $\delta\theta^{(\mathrm{GA})}$ is the same as $\delta\theta^{(\mathrm{OG})}$ with $N$ replaced by $N/k$ and $p_{\psi_\theta}^{(m)}$ replaced by $p_{\psi_\theta^{\mathcal{G}}}^{(m)}$. Therefore the CLT results for $\delta\theta^{(\mathrm{OG})}$ and $\delta\theta^{(\mathrm{GA})}$ are direct consequences of Theorem D.1 by defining $\delta\theta_l^{(\mathrm{OG})}, F_{1l}^{(\mathrm{OG})}, \sigma_l^{(\mathrm{OG})}, \tilde{F}^{(\mathrm{OG})}, \delta\theta_l^{(\mathrm{GA})}, F_{1l}^{(\mathrm{GA})}, \sigma_l^{(\mathrm{GA})}$ and $\tilde{F}^{(\mathrm{GA})}$ as the analogous quantities. We do not state these results for brevity.

## E. Smoothed canonicalization

This appendix expands on the discussion in Section 4.3 and Section 5. We restrict our attention to a diagonal group $\mathbb{G}_{\mathrm{diag}}$ induced by a space group $\mathbb{G} = \mathbb{G}_{\mathrm{sp}}$.

The essential idea behind canonicalization is that, to construct an invariant layer under $\mathbb{G}_{\mathrm{diag}}$, one may perform a "projection" onto a "smallest invariant set" $\Pi \subset \mathbb{R}^{3n}$, called the fundamental region. Formally, $\Pi$ is a fixed set that contains exactly one point from each orbit of $\mathbb{G}_{\mathrm{diag}}$ – called the *representative* point of the orbit – and canonicalization is an $\mathbb{R}^{3n} \to \Pi$ map that brings an input $x$ to its corresponding representative. In the simple case of unit translations and $n = 1$, a standard canonicalization is $x \mapsto x \bmod 1$ with $\Pi = [0, 1)$, as visualized in Fig. 4. A naive canonicalization suffers from boundary discontinuity that violates physical constraints: In the case of $\bmod$, this arises because

$$\lim_{\epsilon \to 0^+} (\epsilon \bmod 1) = 0 \neq 1 = \lim_{\epsilon \to 0^-} (\epsilon \bmod 1) . \tag{8}$$

Resolving this requires introducing smoothing at the boundary. As an example, consider the supercell assumption (Rajagopal et al., 1995; Kittel & McEuen, 2018), which corresponds to separate invariance under translations along each 1d lattice vector. Jastrow factors ((Whitehead et al., 2016); also see (Li et al., 2022) for its implementation in DeepSolid) exactly perform a smoothed version of canonicalization along each 1d lattice vector: These factors take the form $h(x \bmod 1)$, where $x \in \mathbb{R}$ is a suitably rescaled input along one lattice vector and $h$ is a piecewise polynomial chosen such that $x \mapsto h(x \bmod 1)$ is twice continuously differentiable. However, the construction of $h$ is easy here only because the fundamental regions of 1d translations have simple boundaries (two endpoints), and does not generalize well to more complicated symmetries.

For groups such as permutations and rotations, Dym, Lawrence, and Siegel (2024) proposes a smoothing method by taking weighted averages at the boundary. We adapt this method for our $\mathbb{G}_{\text{diag}}$ induced by a space group, and show how it can be achieved by operating with the group $\mathbb{G}$ acting on single electrons. This requires two ingredients:

- **Fundamental region of $\mathbb{G}_{\text{diag}}$.** There are infinitely many choices of fundamental regions, and we shall fix one that is convenient for our canonicalization. Fix $\Pi_0 \subset \mathbb{R}^3$, a fundamental region of $\mathbb{R}^3$ under $\mathbb{G}$. Then $\Pi := \Pi_0 \times (\mathbb{R}^3)^{n-1}$ forms a fundamental region under $\mathbb{G}_{\text{diag}}$, as it is intersected exactly once by any orbit $\{(g(x_1), \ldots, g(x_n)) \mid g \in \mathbb{G}\}$. An advantage of this choice is that, since the boundary of $\Pi$ is completely determined by the boundary of $\Pi_0$ in the space of the first electron, we may perform smoothing directly near the boundary of $\Pi_0$. Also notice that we may equivalently choose the fundamental region $\Pi^{(k)} := (\mathbb{R}^3)^{k-1} \times \Pi_0 \times (\mathbb{R}^3)^{n-k-1}$ for any $1 \le k \le n$.

- **Smoothing factor.** Fix some small $\epsilon > 0$. For each $x \in \mathbb{R}^3$, consider the set of group elements that moves $x$ to be at most $\epsilon$-away from $\Pi_0$:

$$\mathcal{G}_\epsilon(x) := \big\{ g \in \mathbb{G} \,\big|\, d(g(x), \Pi_0) \le \epsilon \big\} \,,$$

where $d$ is some distance preserved by the isometries in $\mathbb{G}$:

$$d(g(x), g(\Pi_0)) = d(x, \Pi_0) \ \text{ for all } g \in \mathbb{G} \text{ and } x \in \mathbb{R}^3 \,.$$

Adapting the weighted averaging idea from Dym et al. (2024), we perform smoothing in the $\epsilon$-neighborhood of $\Pi_0$ by assigning weights to the different group operations: For each $g \in \mathbb{G}$, we define

$$w_\epsilon^g(x) := \frac{\lambda_\epsilon(\, d(g(x), \Pi_0)\,)}{\sum_{g' \in \mathcal{G}_\epsilon(x)} \lambda_\epsilon(\, d(g'(x), \Pi_0)\,)} \ \in \ [0, 1] \,,$$

where $\lambda_\epsilon : \mathbb{R} \to [0, 1]$ is a strictly decreasing function such that $\lambda_\epsilon(w) = 1$ for all $w \le 0$ and $\lambda_\epsilon(w) = 0$ for all $w \ge \epsilon$. If $d(g(x), \Pi_0) \ge \epsilon$, $g$ moves $x$ "too far", and is assigned zero weights. If $d(g(x), \Pi_0) = 0$, $g$ keeps $x$ within the closure of $\Pi_0$, and is assigned a weight of 1. In particular, if $x \in \Pi_0$ is a point of high symmetry and $\epsilon$ is small enough such that $g(x) = x$ for all $g \in \mathcal{G}_\epsilon(x)$, the weight of every $g \in \mathcal{G}_\epsilon(x)$ is $\frac{1}{|\mathcal{G}_\epsilon(x)|}$: Here, all operations that preserve $x$ are assigned equal weights. See Appendix E.1 for explicit choices o $d$ and $\lambda_\epsilon$.

With these tools at hand, we define the smoothed canonicalization (SC) of an unsymmetrized wavefunction $f_\theta$ as

$$\psi_{\theta;\epsilon}^{(\text{SC})}(\mathbf{x}) := \frac{1}{n} \sum_{k=1}^n \psi_{\theta;\epsilon}^{(\text{SC};k)}(\mathbf{x}) \,, \qquad \text{where} \quad \psi_{\theta;\epsilon}^{(\text{SC};k)}(\mathbf{x}) := \sum_{g \in \mathcal{G}_\epsilon(x_k)} w_\epsilon^g(x_k) \, \psi_\theta(g(\mathbf{x})) \,. \tag{9}$$

Appendix E.2 illustrates (9) through an one-electron example. The next result guarantees the validity of $\psi_{\theta;\epsilon}^{(\text{SC})}(\mathbf{x}) = \frac{1}{n} \sum_{k=1}^n \psi_{\theta;\epsilon}^{(\text{SC};k)}(\mathbf{x})$ as a symmetrized wavefunction: Theorem E.1(i) proves diagonal invariance, (ii) proves anti-symmetry and (iii) ensures smoothness under appropriate choices of $\lambda_\epsilon$ and $d$.

**Theorem E.1.** *Fix $\theta \in \mathbb{R}^q$ and $\epsilon > 0$. The following properties hold for $\psi_{\theta;\epsilon}^{(\text{SC})}$ and $\psi_{\theta;\epsilon}^{(\text{SC};k)}$ for all $1 \le k \le n$:*

*(i) $\psi_{\theta;\epsilon}^{(\text{SC};k)}(g(\mathbf{x})) = \psi_{\theta;\epsilon}^{(\text{SC};k)}(\mathbf{x})$ for all $g \in \mathbb{G}$ and $\mathbf{x} \in \mathbb{R}^{3n}$;*

*(ii) Let the spin-dependence in $\psi_\theta$ be explicit, i.e. we view $\psi_\theta$ as an $(\mathbb{R}^3 \times \{\uparrow, \downarrow\})^n \to \mathbb{C}$ function. If $\psi_\theta$ is anti-symmetric with respect to permutations of $(x_i, \sigma_i)$, then so is $\psi_{\theta;\epsilon}^{(\text{SC})}$;*

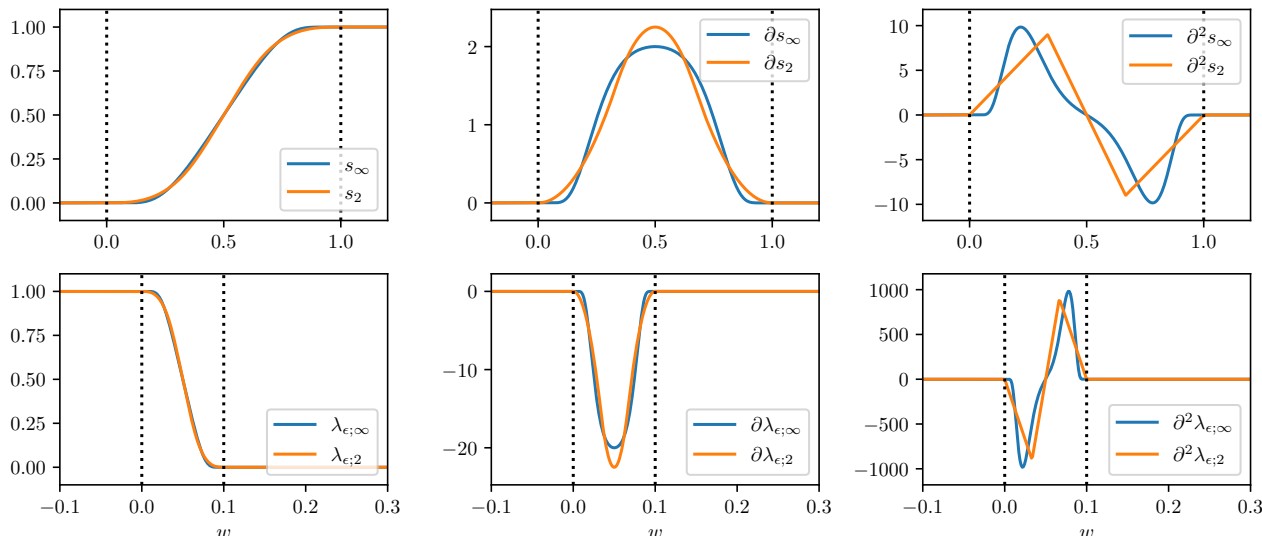

*Figure 10.* Illustration of different choices of step functions $s$, the corresponding $\lambda_\epsilon$'s and their first two derivatives. $\epsilon = 0.1$ above.

*(iii) Suppose $\lambda_\epsilon$ and $d(\bullet, g(\Pi_0))$ are $p$-times continuously differentiable for all $g \in \mathbb{G}$, and that $\psi_\theta$ is $p$-times continuously differentiable at $g(\mathbf{x}) \in \mathbb{R}^{3n}$ for all $g \in \mathbb{G}$. Then $\psi_{\theta;\epsilon}^{(\mathrm{SC})}$ is also $p$-times continuously differentiable at $\mathbf{x}$. Moreover for $0 \leq q \leq p$, the $q$-th derivative of $\psi_{\theta;\epsilon}^{(\mathrm{SC})}$ at $\mathbf{x} = (x_1, \ldots, x_n)$ can be computed as*

$$\nabla^q \psi_{\theta;\epsilon}^{(\mathrm{SC})}(x_1, \ldots, x_n) = \frac{1}{n} \sum_{k=1}^n \sum_{g \in \mathcal{G}_\epsilon(x_k)} \nabla^q \psi_{\theta,\epsilon;g,x_k}^{(\mathrm{SC};k)}(\mathbf{x}),$$

*where we have defined, for $1 \leq k \leq n$, $g \in \mathbb{G}$ and $x, y_1, \ldots, y_n \in \mathbb{R}^3$,*

$$\psi_{\theta,\epsilon;g,x}^{(\mathrm{SC};k)}(y_1, \ldots, y_n) := \frac{\lambda_\epsilon\big(d(y_k, g(\Pi_0))\big)}{\sum_{g' \in \mathcal{G}_\epsilon(x)} \lambda_\epsilon\big(d(y_k, g'(\Pi_0))\big)} \psi_\theta\big(g^{-1}(y_1), \ldots, g^{-1}(y_n)\big).$$

**Computational cost compared to DA and GA.** As mentioned in Section 4.3, SC suffers from a similar computational issue as DA and GA as it involves averaging. However, the amount of averaging SC requires is different. The additional averaging over $n$ fundamental regions is crucial for ensuring anti-symmetry, which is made precise in Theorem E.1 and its proof in Appendix H. Averaging is also required over the set $\mathcal{G}_\epsilon$, which necessarily includes all point group operations and unit translations of $\mathbb{G}$ (note that $\mathcal{G}_\epsilon$ is not a group). For a fair comparison, compare SC to DA and GA that average over the group $\mathbb{G}$. For small systems with $1 \times 1 \times 1$ supercell, $\mathcal{G}_\epsilon$ is strictly larger than the point group $\mathbb{G}$, and the amount of averaging required by SC, $n|\mathcal{G}_\epsilon|$, is strictly larger. For large systems, the cost of SC scales differently from DA and GA: While the cost of DA and GA increase as a function of the increasing group size $|\mathbb{G}|$, for SC, $\mathcal{G}_\epsilon$ is fixed and the cost increases as a function of the number of electrons $n$.

**Post hoc smoothed canonicalization (PC).** As discussed in Section 5, we may perform canonicalization only at evaluation. However, there is an inherent tradeoff in the choice of $\epsilon$: If $\epsilon$ is large, averaging happens in a large neighborhood of the boundary, which can cause the performance of $\psi_{\hat{\theta};\epsilon}^{(\mathrm{SC})}$ to deviate significantly from the well-trained $\psi_{\hat{\theta}}^{(\mathrm{OG})}$ (see Appendix E.2). If $\epsilon$ is small, there may be a blowup in the derivatives of the weights via the smoothing function:

**Lemma E.2.** *If $\lambda_\epsilon$ is twice continuously differentiable, there are $y_1, y_2 \in [0, \epsilon]$ s.t. $\partial \lambda_\epsilon(y_1) = \epsilon^{-1}$, $\partial^2 \lambda_\epsilon(y_2) = \epsilon^{-2}$.*

Since the local energy calculation involves $\partial^2 \lambda_\epsilon$, this can lead to an $\epsilon^{-2}$ blowup of local energy in a region of size $O(\epsilon)$, resulting in larger fluctuations of local energy. This inflation in the variance is clearly visible in Table 2.

### E.1. Choices of $\lambda_\epsilon$ and $d$

To obtain a $p$-times continuously differentiable $\psi_{\theta,\epsilon}^{(\mathrm{SC})}$, Theorem E.1(iii) requires a smoothing function $\lambda_\epsilon$ and a distance function $d(\bullet, g(\Pi_0))$ that are $p$-times continuous differentiable for all $g \in \mathbb{G}$.

**Construction of $\lambda_\epsilon$.** Since $\lambda_\epsilon : \mathbb{R} \to [0,1]$ is required to be strictly decreasing with $\lambda_\epsilon(w) = 1$ for $w \leq 0$ and $\lambda(w) = 0$ for all $w \geq \epsilon$, the function

$$s(w) \;:=\; \lambda_\epsilon\big(\epsilon(1-w)\big)$$

is a $p$-times continuously differentiable approximation of a step function: $s$ is strictly increasing, $s(w) = 0$ for all $w \leq 0$ and $s(w) = 1$ for all $w \geq 1$. Many choices are available for such a function: To have an infinitely differentiable $\lambda_\epsilon$, one may consider the smoothed step function

$$s_\infty(w) \;:=\; \frac{\phi(w)}{\phi(w) + \phi(1-w)} \qquad \text{where } \phi(w) \;:=\; \begin{cases} \exp(-w^{-1}) & \text{if } w > 0, \\ 0 & \text{if } w \leq 0, \end{cases}$$

and use the relationship $\lambda_\epsilon(w) = s\big(1 - w/\epsilon\big)$ to obtain

$$\lambda_{\epsilon;\infty}(w) \;:=\; \frac{\phi(1 - w/\epsilon)}{\phi(1 - w/\epsilon) + \phi(w/\epsilon)} \;.$$

As we only need to evaluate the Hamiltonian, twice continuous differentiability typically suffices for our problem. Another choice of the step function $s$ and the corresponding $\lambda_\epsilon$ are the degree-three polynomial splines

$$s_2(w) := \begin{cases} 0 & \text{if } w \leq 0, \\ \frac{9w^3}{2} & \text{if } w \in (0, \frac{1}{3}], \\ -\frac{9(1-w)^3}{2} + \frac{(2-3w)^3}{2} + 1 & \text{if } w \in (\frac{1}{3}, \frac{2}{3}], \\ -\frac{9(1-w)^3}{2} + 1 & \text{if } w \in (\frac{2}{3}, 1], \\ 1 & \text{if } w > 1, \end{cases} \qquad \lambda_{\epsilon;2}(w) := \begin{cases} 1 & \text{if } w \leq 0, \\ 1 - \frac{9w^3}{2\epsilon^3} & \text{if } w \in (0, \frac{\epsilon}{3}], \\ -\frac{9w^3}{2\epsilon^3} + \frac{(3w-\epsilon)^3}{2\epsilon^3} + 1 & \text{if } w \in (\frac{\epsilon}{3}, \frac{2\epsilon}{3}], \\ \frac{9(\epsilon-w)^3}{2\epsilon^3} & \text{if } w \in (\frac{2\epsilon}{3}, \epsilon], \\ 0 & \text{if } w > \epsilon, \end{cases}$$

Fig. 10 plots $s_\infty$, $s_2$, $\lambda_{\epsilon;\infty}$, $\lambda_{\epsilon;2}$ and their derivatives. Note that the derivative plots for $\lambda_\epsilon$'s verify Lemma E.2. To achieve general $p$-times continuous differentiability, we refer readers to the many constructions of such step functions available in the statistics literature, e.g. Theorem 3.3 of Chen et al. (2011) or Lemma 34 of Huang et al. (2023). In particular, our $s_2$ is related to $h_{m;\tau;\delta}$ constructed in Lemma 34 of Huang et al. (2023) via $s_2(w) = h_{2;1;1}(w)$; one can similarly take their $h_{p;1;1}$ to obtain a $p$-times continuously differentiable $s_p$ and obtain the corresponding $\lambda_{\epsilon;p}(w) := 1 - s_p(w/\epsilon)$.

**Constructing $d(\bullet, g(\Pi_0))$ via the step function $s$.** A $p$-times continuously differentiable step function $s$, as discussed above, induces a $p$-times continuously differentiable approximation of $\max\{\bullet, 0\}$, given as

$$\tilde{s}(w) \;:=\; w\,s(w) \;.$$

In particular $\tilde{s}(w) = 0$ if and only if $w \leq 0$, and $\tilde{s}(w) = w$ for $w \geq 1$. To utilize $\tilde{s}$ to construct $d(\bullet, g(\Pi_0))$, we notice that each $g(\Pi_0)$ itself is a fundamental region, and its closure is a simplex completely characterized by the relation

$$x \in g(\Pi_0) \;\Leftrightarrow\; \frac{(x - g(c_0))^\top n_l}{\|n_l\|^2} \leq 1 \text{ for } 1 \leq l \leq m,$$

where $c_0$ is the center of $\Pi_0$ and $n_l$ is the normal vector starting from $c_0$ that is normal to the $l$-th face of $\Pi_0$. This allows us to define

$$d(x, g(\Pi_0)) \;:=\; \sum_{l=1}^m \left( \tilde{s}\Big( \frac{(x - g(c_0))^\top n_l}{\|n_l\|^2} - 1 \Big) \right)^2 \;.$$

This distance function satisfies that

- $d(g(x), g(\Pi_0)) = d(x, \Pi_0)$ for all $g \in \mathbb{G}$, which can be verified by noting that $g(\bullet) = A(\bullet) + b$ for an orthogonal matrix $A \in \mathbb{R}^{3 \times 3}$ and a translation vector $b \in \mathbb{R}^3$;

- $d(x, \Pi) = 0$ if and only if $x \in \Pi$, and the map $x \mapsto d(x, \Pi)$ is $p$-times continuously differentiable.

To visualize this distance function in the case of a 1d system, see Fig. 14.

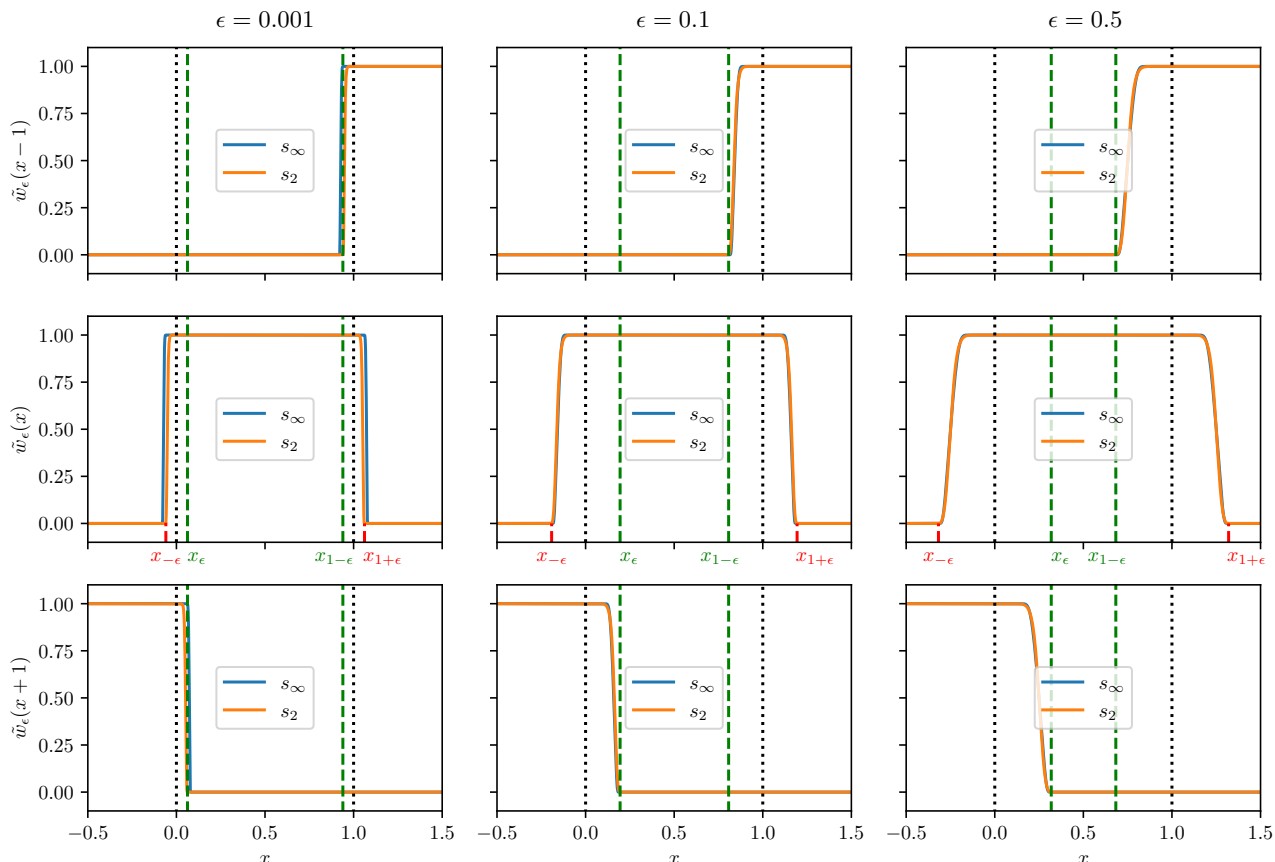

*Figure 11.* Plots of $\tilde{w}_\epsilon(x-1)$, $\tilde{w}_\epsilon(x)$ and $\tilde{w}_\epsilon(x+1)$ defined via either $s_\infty$ or $s_2$ as the smoothed step function used in $\lambda_\epsilon$ and $d$, and for different values of $\epsilon$.

## E.2. An one-electron example of $\psi_{\theta;\epsilon}^{(\mathrm{SC})}$

For simplicity, we first illustrate $\psi_{\theta;\epsilon}^{(\mathrm{SC})}$ for a single 1d-electron system with unit translation invariance. In this case, the group $\mathbb{G}$ is generated by translations of length 1, and the fundamental region is $\Pi_0 = [0, 1)$, as illustrated in Fig. 13. As discussed at (8), a standard canonicalization is to take $x \mapsto x \bmod 1$, which suffers from discontinuity at the boundary. Our proposed smoothed canonicalization in this case becomes

$$\psi_{\theta;\epsilon}^{(\mathrm{SC})}(x) \;=\; \sum_{t \in \mathbb{Z} \text{ s.t. } d(x+t,[0,1))} \frac{\lambda_\epsilon\big(d(x+t,[0,1))\big)}{\sum_{t' \in \mathbb{Z} \text{ s.t. } d(x+t',[0,1))} \lambda_\epsilon\big(d(x+t',[0,1))\big)} \, \psi_\theta(x+t) \, .$$

Choosing the distance function $d(\bullet, [0, 1))$ as in Appendix E.1, we have

$$d(x, [0, 1)) \;=\; \tilde{s}(2(x-1)) + \tilde{s}(-2x) \qquad\qquad \text{where} \qquad\qquad \tilde{s}(w) \;=\; ws(w)$$

and $s$ is a $p$-times differentiable approximation of a step function. Fig. 14 plots $d(x, [0, 1))$ and its derivatives under $s = s_\infty$ and $s = s_2$ from Appendix E.1, and verifies that $d(x, [0, 1))$ is twice continuously differentiable under either case.

Notice by construction that $\psi_{\theta;\epsilon}^{(\mathrm{SC})}$ is 1-periodic (also see Theorem E.1 for the proof in the general case), so it suffices to consider the value of $\psi_{\theta;\epsilon}^{(\mathrm{SC})}(x)$ within the interval $x \in [0, 1]$. Clearly $d(x, [0, 1)) = 0$. By construction of the distance function, the only possible translations $t$ with non-zero (unnormalized) weight, defined as

$$\tilde{w}_\epsilon(x+t) \;:=\; \lambda_\epsilon\big(d(x+t, [0, 1))\big) \, ,$$

are $t = \pm 1$. In other words, for $x \in [0, 1]$, we can express

$$\psi_{\theta;\epsilon}^{(\mathrm{SC})}(x) \;=\; \frac{\tilde{w}_\epsilon(x-1)}{S_\epsilon(x)} \, \psi_\theta(x-1) + \frac{\tilde{w}_\epsilon(x)}{S_\epsilon(x)} \, \psi_\theta(x) + \frac{\tilde{w}_\epsilon(x+1)}{S_\epsilon(x)} \, \psi_\theta(x+1) \, ,$$

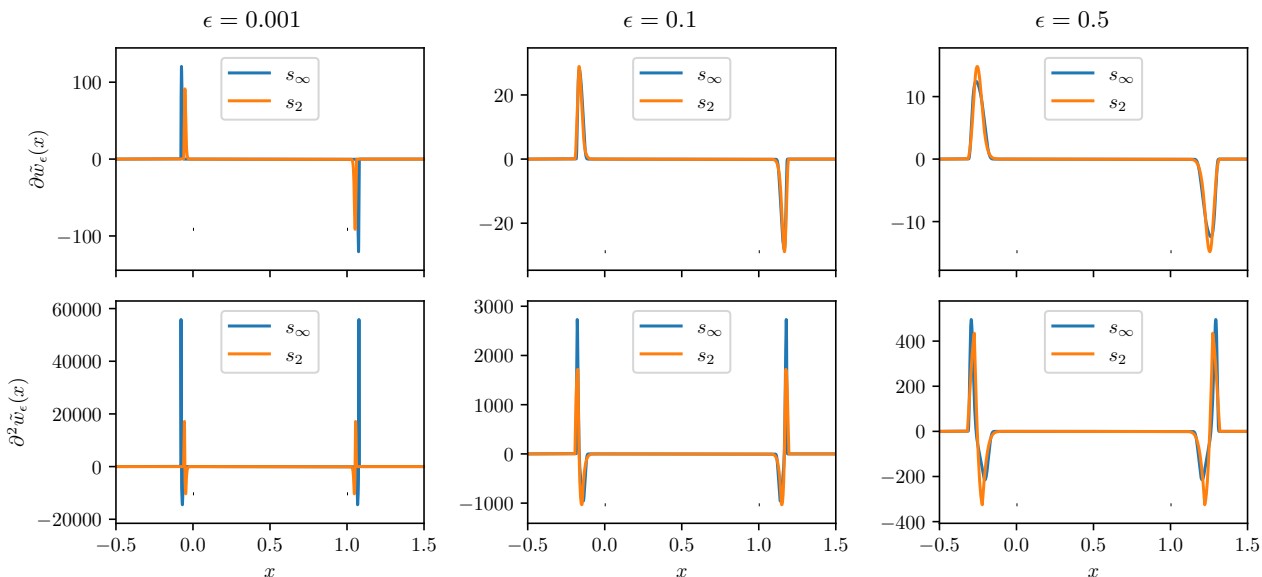

*Figure 12.* Plots of derivatives of $\tilde{w}_\epsilon(x)$ for different values of $\epsilon$.

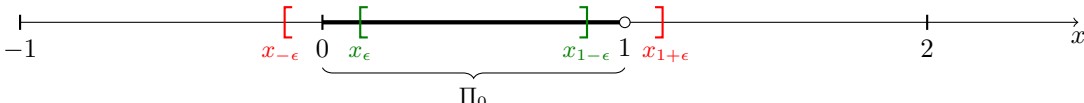

*Figure 13.* Illustration of smoothed canonicalization $\psi_{\theta;\epsilon}^{(\text{SC})}$ for a single 1d electron

where we have denoted the normalizing term as $S_\epsilon(x) := \tilde{w}_\epsilon(x-1) + \tilde{w}_\epsilon(x) + \tilde{w}_\epsilon(x+1)$. Fig. 11 plots the values of the three unnormalized weights as $x$ varies, under different choices of $\epsilon$. Several observations follow:

- For $x \in [0, 1]$, we always have $\tilde{w}_\epsilon(x) = 1$, whereas $\tilde{w}_\epsilon(x-1)$ is non-zero only for $x$ close to 1, and $\tilde{w}_\epsilon(x+1)$ is non-zero only for $x$ close to 0. In particular, $\psi_{\theta;\epsilon}^{(\text{SC})}(x)$ is exactly the original wavefunction $\psi_\theta(x)$ when $x$ is far away from the edge of the interval, whereas an weighted average is taken with $\psi_\theta(x+1)$ if $x$ is close to 0, and with $\psi_\theta(x-1)$ if $x$ is close to 1.

- At $x = 0$ and $x = 1$, $\psi_{\theta;\epsilon}^{(\text{SC})}(x)$ is exactly the arithmetic mean of $\psi_\theta(0)$ and $\psi_\theta(1)$. Indeed, smooth canonicalization allows the wavefunction to smoothly interpolate to an average exactly at the boundary of the fundamental region.

**Effect on the performance of $\psi_{\theta;\epsilon}^{(\text{SC})}$ as $\epsilon$ increases.** Compare the exact canonicalization $\psi_\theta(x \bmod 1)$ with $\psi_{\theta;\epsilon}^{(\text{SC})}$: In the former case, the wavefunction value depends only on the values of $\psi_\theta$ within the fundamental region $\Pi_0 = [0, 1)$, whereas in the latter case, $\psi_{\theta;\epsilon}^{(\text{SC})}$ depends on $\psi_\theta$ evaluated on a slightly larger set $\{x \in \mathbb{R} \mid d(x, [0, 1)) \le \epsilon\}$, represented as the interval $[x_{-\epsilon}, x_{1+\epsilon}]$ in both Fig. 13 and Fig. 11. Meanwhile, $\psi_\theta(x \bmod 1) = \psi_\theta(x)$ for all $x \in (0, 1)$, whereas $\psi_{\theta;\epsilon}^{(\text{SC})}(x) = \psi_\theta(x)$ only for a slightly smaller set $x \in [x_\epsilon, x_{1-\epsilon}]$, labelled in Fig. 13 and Fig. 11. Notably for post hoc canonicalization, as $\epsilon$ increases, the performance of $\psi_{\theta;\epsilon}^{(\text{SC})}$ relies on $\psi_\theta$ to be a well-trained on a larger and larger region, and $\psi_{\theta;\epsilon}^{(\text{SC})}$ only recovers the trained $\psi_\theta$ on a smaller and smaller region, thus retaining fewer benefits from training.

**Gradient blowup as $\epsilon$ decreases.** Lemma E.2 and Fig. 10 both show that the derivatives of $\lambda_\epsilon$ blow up as $\epsilon$ gets small. This also leads to a blowup in derivatives of $\tilde{w}_\epsilon(x)$ and hence those of $\psi_\theta^{(\text{SC})}$ near the boundary. Since the derivatives of $\tilde{w}_\epsilon(x)$ involve products of derivatives of $\lambda_\epsilon$ and derivatives of $d$ by the chain rule, the difference in the magnitude of blowup across $s = s_2$ and $s = s_\infty$ is more pronounced, as illustrated in Fig. 12: defining the weight function $\tilde{w}_\epsilon$ via $s_\infty$ leads to a higher degree of smoothness, but at the cost of a larger blowup in the derivative.

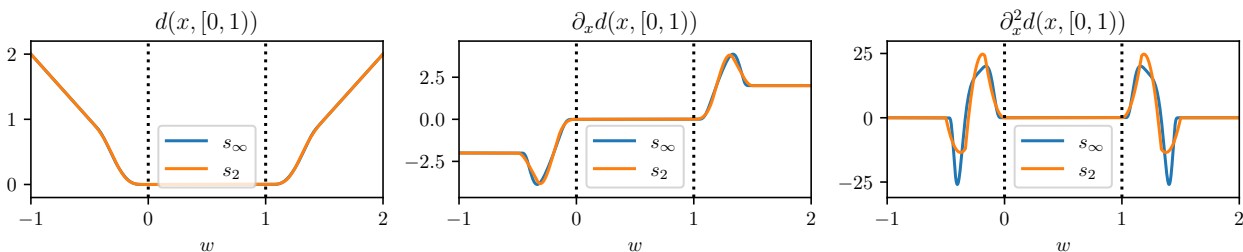

*Figure 14.* Plots of $d(x, [0, 1))$ and its derivatives under different choices of the step function $s$.

# F. Proof of Fact 2.1

Fix $g \in \mathbb{G}$ with its action on $x \in \mathbb{R}^3$ represented by $g(x) = Ax + b$. Given that $(\psi, E)$ solves (1), we seek to show that under the stated conditions, $\psi_g(\mathbf{x}) := \psi(g(\mathbf{x}))$ is also an anti-symmetric solution to (1) with respect to the same energy $E$. For $\mathbf{x} \in \mathbb{R}^{3n}$ and $g \in \mathbb{G}$, denote $\mathbf{x}_g = g(\mathbf{x})$. Then for every $\mathbf{x} \in \mathbb{R}^{3n}$,

$$
\begin{aligned}
\left( -\frac{1}{2}\nabla^2 + V(\mathbf{x}) \right) \psi_g(\mathbf{x}) &= \left( -\frac{1}{2}\nabla^2 + V(\mathbf{x}) \right) \psi(Ax_1 + b, \dots, Ax_n + b) \\
&= -\frac{1}{2}(A^\top A)\nabla^2 \psi(\mathbf{x}_g) + V(\mathbf{x})\psi(\mathbf{x}_g) \\
&\overset{(a)}{=} -\frac{1}{2}\nabla^2 \psi(\mathbf{x}_g) + V(\mathbf{x}_g)\psi(\mathbf{x}_g) \overset{(b)}{=} E\psi(\mathbf{x}_g) = E\psi_g(\mathbf{x}) .
\end{aligned}
$$

In $(a)$, we have used the $\mathbb{G}_{\mathrm{diag}}$-invariance of $V$; in $(b)$, we used that $(\psi, E)$ solve (1). Since the above holds for all $\mathbf{x} \in \mathbb{R}^{3n}$, we get that $(\psi_g, E)$ also solves the Schrödinger's equation. To verify the anti-symmetric requirement, we write the wavefunction $\psi$ as $\psi(\tilde{\mathbf{x}}) = \psi(\tilde{x}_1, \dots, \tilde{x}_n)$, where each $\tilde{x}_i = (x_i, \sigma_i)$ now additionally depends on the spin $\sigma_i \{\uparrow, \downarrow\}$. Since $\mathbb{G}$ only acts on the spatial position in $\mathbb{R}^3$ and leaves the spins invariant, we can WLOG express the $g$-transformed version of $\tilde{x}_i$ as $g(\tilde{x}_i) = (g(x_i), \sigma_i)$. Moreover, since $\sigma \in P_n$ commutes with the diagonal action of $g \in \mathbb{G}$ on $(\mathbb{R}^3 \times \{\uparrow, \downarrow\})^n$,

$$
\psi_g(\sigma(\tilde{\mathbf{x}})) = \psi(g \circ \sigma(\tilde{\mathbf{x}})) = \psi(\sigma \circ g(\tilde{\mathbf{x}})) \overset{(c)}{=} \mathrm{sgn}(\sigma)\,\psi(g(\tilde{\mathbf{x}})) = \mathrm{sgn}(\sigma)\,\psi_g(\tilde{\mathbf{x}}) .
$$

In $(c)$, we have used the anti-symmetry of $\psi$. This proves that $(\psi_g, E)$ solves (1) with the correct anti-symmetric requirement. To prove the theorem statement, we see that $\psi^{\mathbb{G}}$ is a linear combination of finitely many eigenfunctions $(\psi_g)_{g \in \mathcal{G}}$ with the same eigenvalue $E$ and therefore yields an anti-symmetric solution with the same energy $E$. $\qquad\square$

# G. Proofs for data augmentation and group-averaging

## G.1. Proof of Proposition 4.1

We first compute the difference in expectation as

$$
\begin{aligned}
\|\mathbb{E}[\delta\theta^{(\mathrm{DA})}] - \mathbb{E}[\delta\theta^{(\mathrm{OG})}]\| &= \left\| \mathbb{E}\left[ \frac{1}{N} \sum_{i \le N/k} \sum_{j \le k} F_{\mathbf{g}_{i,j}(\mathbf{X}_i); \psi_\theta} \right] - \mathbb{E}\left[ \frac{1}{N} \sum_{i \le N} F_{\mathbf{X}_i; \psi_\theta} \right] \right\| \\
&= \|\mathbb{E}[F_{\mathbf{g}_{1,1}(\mathbf{X}_1); \psi_\theta}] - \mathbb{E}[F_{\mathbf{X}_1; \psi_\theta}]\| = \left\| \mathbb{E}_{\mathbf{X} \sim p^{(m)}_{\psi_\theta; \mathrm{DA}}}[F_{\mathbf{X}; \psi_\theta}] - \mathbb{E}_{\mathbf{Y} \sim p^{(m)}_{\psi_\theta}}[F_{\mathbf{Y}; \psi_\theta}] \right\| .
\end{aligned}
$$

In the last line, we used linearity of expectation, the fact that $\mathbf{g}_{i,j}(\mathbf{X}_i)$'s are identically distributed and that $\mathbf{X}_i$'s are identically distributed. Recall that $\left\{ \mathbf{x} \mapsto \left( F_{\mathbf{x}; \psi_\theta}, F^{\otimes 2}_{\mathbf{x}; \psi_\theta} \right) \mid \theta \in \mathbb{R}^q \right\} \subseteq \mathcal{F}$ and $d_{\mathcal{F}}(p, q) = \sup_{f \in \mathcal{F}} \|\mathbb{E}_{\mathbf{X} \sim p}[f(\mathbf{X})] - \mathbb{E}_{\mathbf{Y} \sim q}[f(\mathbf{Y})]\|$. This implies that

$$
\|\mathbb{E}[\delta\theta^{(\mathrm{DA})}] - \mathbb{E}[\delta\theta^{(\mathrm{OG})}]\| = \left\| \mathbb{E}_{\mathbf{X} \sim p^{(m)}_{\psi_\theta; \mathrm{DA}}}[F_{\mathbf{X}; \psi_\theta}] - \mathbb{E}_{\mathbf{Y} \sim p^{(m)}_{\psi_\theta}}[F_{\mathbf{Y}; \psi_\theta}] \right\| \le d_{\mathcal{F}}\big( p^{(m)}_{\psi_\theta; \mathrm{DA}}, p^{(m)}_{\psi_\theta} \big),
$$

which proves the first bound. For the second bound, notice that

$$
\begin{aligned}
\mathrm{Var}[\delta\theta^{(\mathrm{DA})}] &= \mathrm{Var}\Big[\frac{1}{N}\sum_{i\leq N/k}\sum_{j\leq k}F_{\mathbf{g}_{i,j}(\mathbf{X}_i);\psi_\theta}\Big]\\
&\stackrel{(a)}{=}\frac{1}{N/k}\mathrm{Var}\Big[\frac{1}{k}\sum_{j\leq k}F_{\mathbf{g}_{1,j}(\mathbf{X}_1);\psi_\theta}\Big]\\
&\stackrel{(b)}{=}\frac{1}{N}\mathrm{Var}[F_{\mathbf{g}_{1,1}(\mathbf{X}_1);\psi_\theta}]+\frac{k-1}{N}\mathrm{Cov}[F_{\mathbf{g}_{1,1}(\mathbf{X}_1);\psi_\theta},F_{\mathbf{g}_{1,2}(\mathbf{X}_1);\psi_\theta}]\\
&\stackrel{(c)}{=}\frac{1}{N}\mathrm{Var}[F_{\mathbf{g}_{1,1}(\mathbf{X}_1);\psi_\theta}]+\frac{k-1}{N}\mathrm{Var}\,\mathbb{E}\big[F_{\mathbf{g}_{1,1}(\mathbf{X}_1);\psi_\theta}\big|\mathbf{X}_1\big]\,.
\end{aligned}
$$

In $(a)$, we have noted that the summands are i.i.d. across $i\leq N/k$; in $(b)$, we have computed the variance of the sum explicitly by expanding the expectation of a double-sum; in $(c)$, we have applied the law of total covariance to obtain that

$$
\begin{aligned}
&\mathrm{Cov}[F_{\mathbf{g}_{1,1}(\mathbf{X}_1);\psi_\theta},F_{\mathbf{g}_{1,2}(\mathbf{X}_1);\psi_\theta}]\\
&=\underbrace{\mathrm{Cov}\big[\mathbb{E}[F_{\mathbf{g}_{1,1}(\mathbf{X}_1);\psi_\theta}|\mathbf{X}_1],\,\mathbb{E}[F_{\mathbf{g}_{1,2}(\mathbf{X}_1);\psi_\theta}|\mathbf{X}_1]\big]}_{=\,\mathrm{Var}\,\mathbb{E}[F_{\mathbf{g}_{1,1}(\mathbf{X}_1);\psi_\theta}|\mathbf{X}_1]}+\underbrace{\mathbb{E}\,\mathrm{Cov}\big[F_{\mathbf{g}_{1,1}(\mathbf{X}_1);\psi_\theta}\,,\,F_{\mathbf{g}_{1,2}(\mathbf{X}_1);\psi_\theta}\,\big|\,\mathbf{X}_1\big]}_{=0}\,.
\end{aligned}
$$

Meanwhile, the same calculation with $k=1$ and $\mathbf{g}_{1,1}$ replaced by identity gives

$$
\mathrm{Var}[\delta\theta^{(\mathrm{OG})}] = \frac{1}{N}\mathrm{Var}[F_{\mathbf{X}_1;\psi_\theta}]\,.
$$

Taking a difference and applying the triangle inequality twice, we have

$$
\begin{aligned}
&\Big\|\mathrm{Var}[\delta\theta^{(\mathrm{DA})}]-\mathrm{Var}[\delta\theta^{(\mathrm{OG})}]-\frac{k-1}{N}\mathrm{Var}\,\mathbb{E}\big[F_{\mathbf{g}_{1,1}(\mathbf{X}_1);\psi_\theta}\big|\mathbf{X}_1\big]\Big\| = \frac{1}{N}\big\|\mathrm{Var}[F_{\mathbf{g}_{1,1}(\mathbf{X}_1);\psi_\theta}]-\mathrm{Var}[F_{\mathbf{X}_1;\psi_\theta}]\big\|\\
&\leq \frac{1}{N}\big\|\mathbb{E}[F^{\otimes 2}_{\mathbf{g}_{1,1}(\mathbf{X}_1);\psi_\theta}]-\mathbb{E}[F^{\otimes 2}_{\mathbf{X}_1;\psi_\theta}]\big\|+\frac{1}{N}\big\|\mathbb{E}[F_{\mathbf{g}_{1,1}(\mathbf{X}_1);\psi_\theta}]^{\otimes 2}-\mathbb{E}[F_{\mathbf{X}_1;\psi_\theta}]^{\otimes 2}\big\|\\
&\leq \frac{1}{N}\big\|\mathbb{E}[F^{\otimes 2}_{\mathbf{g}_{1,1}(\mathbf{X}_1);\psi_\theta}]-\mathbb{E}[F^{\otimes 2}_{\mathbf{X}_1;\psi_\theta}]\big\|+\frac{1}{N}\big\|\mathbb{E}[F_{\mathbf{g}_{1,1}(\mathbf{X}_1);\psi_\theta}]+\mathbb{E}[F_{\mathbf{X}_1;\psi_\theta}]\big\|\,\big\|\mathbb{E}[F_{\mathbf{g}_{1,1}(\mathbf{X}_1);\psi_\theta}]-\mathbb{E}[F_{\mathbf{X}_1;\psi_\theta}]\big\|\\
&\leq \frac{d_{\mathcal{F}}(p^{(m)}_{\psi_\theta;\mathrm{DA}},p^{(m)}_{\psi_\theta})}{N}+\frac{1}{N}\big(2\big\|\mathbb{E}[F_{\mathbf{X}_1;\psi_\theta}]\big\|+d_{\mathcal{F}}(p^{(m)}_{\psi_\theta;\mathrm{DA}},p^{(m)}_{\psi_\theta})\big)\times d_{\mathcal{F}}(p^{(m)}_{\psi_\theta;\mathrm{DA}},p^{(m)}_{\psi_\theta})\\
&=\frac{1+2\|\mathbb{E}[\delta\theta^{(\mathrm{OG})}]\|+d_{\mathcal{F}}(p^{(m)}_{\psi_\theta;\mathrm{DA}},p^{(m)}_{\psi_\theta})}{N}\times d_{\mathcal{F}}(p^{(m)}_{\psi_\theta;\mathrm{DA}},p^{(m)}_{\psi_\theta})\,.
\end{aligned}\tag{10}
$$

This proves the second bound. In the case when $\mathbf{g}(\mathbf{X}_1)\stackrel{d}{=}\mathbf{X}_1$ for all $\mathbf{g}\in\mathbb{G}_{\mathrm{diag}}$, we have $p^{(m)}_{\psi_\theta;\mathrm{DA}}=p^{(m)}_{\psi_\theta}$ and therefore the bounds above all evaluate to zero. In this case we have $\mathbb{E}[\delta\theta^{(\mathrm{DA})}]=\mathbb{E}[\delta\theta^{(\mathrm{OG})}]$ and

$$
\mathrm{Var}[\delta\theta^{(\mathrm{DA})}]-\mathrm{Var}[\delta\theta^{(\mathrm{OG})}] = \frac{k-1}{N}\mathrm{Var}\,\mathbb{E}\big[F_{\mathbf{g}_{1,1}(\mathbf{X}_1);\psi_\theta}\big|\mathbf{X}_1\big]\,,
$$

which is positive semi-definite. $\qquad\square$

### G.2. Proof of Lemma 4.2

By construction, $\delta\theta^{(\mathrm{GA})}=\frac{1}{N/k}\sum_{i\leq N/k}F_{\mathbf{X}^{\mathcal{G}}_i;\psi^{\mathcal{G}}_\theta}$ is a size-$N/k$ empirical average of i.i.d. quantities. The mean and variance formulas thus follows directly from a standard computation:

$$
\mathbb{E}[\delta\theta^{(\mathrm{GA})}] = \mathbb{E}[F_{\mathbf{X}^{\mathcal{G}}_1;\psi^{\mathcal{G}}_\theta}]\qquad\text{and}\qquad \mathrm{Var}[\delta\theta^{(\mathrm{GA})}] = \frac{\mathrm{Var}[F_{\mathbf{X}^{\mathcal{G}}_1;\psi^{\mathcal{G}}_\theta}]}{N/k}\,.
$$

### G.3. Proof of Theorem D.1

To prove the coordinate-wise bound, fix $l\leq p$. Note that if $\sigma^{(\mathrm{DA})}_l=0$, then $\delta\theta^{(\mathrm{DA})}_{1l}=\mathbb{E}[\delta\theta^{(\mathrm{DA})}_{1l}]$ with probability 1, implying that the distribution difference is zero and hence the bound is satisfied. In the case $\sigma^{(\mathrm{DA})}_l>0$, $\delta\theta^{(\mathrm{DA})}_{1l}=\frac{1}{N/k}\sum_{i\leq N/k}F^{(\mathrm{DA})}_{il}$ is an average of i.i.d. univariate random variables with positive variance. By renormalizing $t$ and

applying the Berry-Esséen theorem (see Theorem 3.7 of Chen et al. (2011) or Shevtsova (2013) for the version with a tight constant $C_1 = 0.469$) applied to $\frac{1}{N/k} \sum_{i \leq N/k} F_{il}$, we get that

$$\sup_{t \in \mathbb{R}} \left| \mathbb{P}\big( \delta\theta_{1l}^{(DA)} \leq t \big) - \mathbb{P}\Big( \mathbb{E}\big[\delta\theta_{1l}^{(DA)}\big] + (\text{Var}[\delta\theta_{1l}^{(DA)}])^{1/2} Z_l \leq t \Big) \right|$$

$$= \sup_{t \in \mathbb{R}} \left| \mathbb{P}\Big( \frac{1}{N/k} \sum_{i \leq N/k} \big( F_{il}^{(DA)} - \mathbb{E}\big[F_{il}^{(DA)}\big] \big) \leq t \Big) - \mathbb{P}\Big( \frac{\sigma_l^{(DA)}}{\sqrt{N/k}} Z_l \leq t \Big) \right| \leq \frac{C_1 \, \mathbb{E}|F_{1l}^{(DA)}|^3}{\sqrt{N/k} \, (\sigma_l^{(DA)})^3} \, .$$

This proves the first set of bounds. To prove the second set of bounds, we first denote the mean-zero variable $\bar{F}_{il} := -F_{il}^{(DA)} + \mathbb{E}[F_{il}^{(DA)}]$, and let $(\bar{Z}_{11}, \ldots, \bar{Z}_{np})$ be an $\mathbb{R}^{np}$-valued Gaussian vector with the same mean and variance as $(\bar{F}_{11}, \ldots, \bar{F}_{np})$. The difference in distribution function can be re-expressed as

$$\sup_{t \in \mathbb{R}} \left| \mathbb{P}\Big( \max_{l \leq p} \big| \delta\theta_{1l}^{(DA)} - \mathbb{E}\big[\delta\theta_{1l}^{(DA)}\big] \big| \leq t \Big) - \mathbb{P}\Big( \max_{l \leq p} \big| (\text{Var}[\delta\theta_{1l}^{(DA)}])^{1/2} Z_l \big| \leq t \Big) \right|$$

$$= \sup_{t \in \mathbb{R}} \left| \mathbb{P}\Big( \max_{l \leq p} \Big| \frac{1}{\sqrt{N/k}} \sum_{i \leq N/k} \bar{F}_{il} \Big| \leq t \Big) - \mathbb{P}\Big( \max_{l \leq p} \Big| \frac{1}{\sqrt{N/k}} \sum_{i \leq N/k} \bar{Z}_{il} \Big| \leq t \Big) \right|, \tag{11}$$

where we have used the definition of $\theta_1^{(DA)}$ and also replaced $t$ by $t/\sqrt{Nk}$. Note that $\bar{F}_{il}$'s are i.i.d. mean-zero across $1 \leq i \leq N/k$ and $\bar{Z}_{il}$'s are i.i.d. mean-zero across $1 \leq i \leq N/k$. As before, if $\sigma_l^{(DA)} = 0$ for all $1 \leq l \leq p$, the two random variables to be compared are both $0$ with probability $1$ and the distributional difference above evaluates to zero. If there is at least one $l$ such that $\sigma_l^{(DA)} = 0$, we can restrict both maxima above to be over $l \leq p$ such that $\sigma_l^{(DA)} = 0$ and ignore the coordinates with zero variance. As such, we can WLOG assume that $\sigma_l^{(DA)} > 0$ for all $l \leq p$. Now write

$$\tilde{F}_i := (\sigma_1^{-1} \bar{F}_{i1}, \ldots, \sigma_p^{-1} \bar{F}_{ip}), \qquad \tilde{Z}_i := (\sigma_1^{-1} \bar{Z}_{i1}, \ldots, \sigma_p^{-1} \bar{Z}_{ip}),$$

and denote the hyper-rectangular set $\mathcal{A}(t) := [-\sigma_1^{-1} t, +\sigma_1^{-1} t] \times \cdots \times [-\sigma_p^{-1} t, +\sigma_p^{-1} t] \subseteq \mathbb{R}^q$. We can now express the difference above further as

$$(11) = \sup_{t \in \mathbb{R}} \left| \mathbb{P}\Big( \frac{1}{\sqrt{N/k}} \sum_{i \leq N/k} \tilde{F}_i \in \mathcal{A}(t) \Big) - \mathbb{P}\Big( \frac{1}{\sqrt{N/k}} \sum_{i \leq N/k} \tilde{Z}_i \in \mathcal{A}(t) \Big) \right|.$$

This is a difference in distribution functions between a normalized empirical average of $p$-dimensional vectors with zero mean and identity covariance and a standard Gaussian in $\mathbb{R}^q$, measured through a subset of hyperrectangles. In particular, this is the quantity controlled by Chernozhukov et al. (2017): Under the stated moment conditions and applying their Proposition 2.1, we have that for some absolute constant $C_2 > 0$,

$$(11) \leq C_2 \Big( \frac{(\tilde{F}^{(DA)})^2 (\log(pN/k))^7}{N/k} \Big)^{1/6} \, . \qquad \square$$

## H. Proofs for results on canonicalization

### H.1. Proof of Theorem E.1

To prove (i), we fix any $\tilde{g} \in \mathbb{G}$, and WLOG let $k = 1$. Then by the definition of $\psi_{\theta;\epsilon}^{(SC;1)}$,

$$\psi_{\theta;\epsilon}^{(SC;1)}(\tilde{g}(x_1), \ldots, \tilde{g}(x_n)) = \sum_{\substack{g \in \mathbb{G} \text{ s.t.} \\ d(\tilde{g}(x_1), g(\Pi_0)) \leq \epsilon}} \Big( \frac{\lambda_\epsilon\big(d(\tilde{g}(x_1), g(\Pi_0))\big)}{\sum_{\substack{g' \in \mathbb{G} \text{ s.t.} \\ d(\tilde{g}(x_1), g'(\Pi_0)) \leq \epsilon}} \lambda_\epsilon\big(d(\tilde{g}(x_1), g'(\Pi_0))\big)} \times \psi_\theta\big(g^{-1}\tilde{g}(x_1), \ldots, g^{-1}\tilde{g}(x_n)\big) \Big).$$

Relabelling $g$ by $\tilde{g}g$ and $g'$ by $\tilde{g}g'$ in the sums above, and noting that $d(\tilde{g}(x_1), \tilde{g}g(\Pi_0)) = d(x_1, g(\Pi_0))$, we obtain that

$$\psi_{\theta;\epsilon}^{(SC;1)}(\tilde{g}(x_1), \ldots, \tilde{g}(x_n)) = \sum_{\substack{g \in \mathbb{G} \text{ s.t.} \\ d(\tilde{g}(x_1), \tilde{g}g(\Pi_0)) \leq \epsilon}} \Big( \frac{\lambda_\epsilon\big(d(\tilde{g}(x_1), \tilde{g}g(\Pi_0))\big)}{\sum_{\substack{g' \in \mathbb{G} \text{ s.t.} \\ d(\tilde{g}(x_1), \tilde{g}g'(\Pi_0)) \leq \epsilon}} \lambda_\epsilon\big(d(\tilde{g}(x_1), \tilde{g}g'(\Pi_0))\big)}$$

$$\times \psi_\theta\big(g^{-1}\tilde{g}^{-1}\tilde{g}(x_1), \ldots, g^{-1}\tilde{g}^{-1}\tilde{g}(x_n)\big) \Big)$$

$$= \sum_{\substack{g \in \mathbb{G} \text{ s.t.} \\ d(x_1, g(\Pi_0)) \leq \epsilon}} \frac{\lambda_\epsilon\big(d(x_1, g(\Pi_0))\big)}{\sum_{\substack{g' \in \mathbb{G} \text{ s.t.} \\ d(x_1, g'(\Pi_0)) \leq \epsilon}} \lambda\big(d(x_1, g'(\Pi_0))\big)} \psi_\theta\big(g^{-1}(x_1), \ldots, g^{-1}(x_n)\big)$$

$$= \psi_{\theta;\epsilon}^{(SC;1)}(x_1, \ldots, x_n) \, .$$

This proves diagonal $\mathbb{G}$-invariance.

To prove (ii), we shall make the spin-dependence explicit and write $\tilde{x}_i = (x_i, \sigma_i)$ and $g(\tilde{x}_i) = (g(x_i), \sigma_i)$. First consider $\pi$, a transposition that swaps the indices 1 and 2. Then

$$\psi_{\theta;\epsilon}^{(\mathrm{SC})}(\tilde{x}_{\pi(1)}, \ldots, \tilde{x}_{\pi(n)}) = \frac{1}{n} \sum_{k=1}^{n} \psi_{\theta;\epsilon}^{(\mathrm{SC};k)}(\tilde{x}_2, \tilde{x}_1, \tilde{x}_3 \ldots, \tilde{x}_n) .$$

By the anti-symmetry of $\psi_\theta$, we see that for $k \geq 3$,

$$\begin{aligned}
\psi_{\theta;\epsilon}^{(\mathrm{SC};k)}(\tilde{x}_2, \tilde{x}_1, \tilde{x}_3 \ldots, \tilde{x}_n) &= \sum_{g \in \mathcal{G}_\epsilon(x_k)} w_\epsilon^g(x_k) \, \psi_\theta\big(g^{-1}(\tilde{x}_2), g^{-1}(\tilde{x}_1), g^{-1}(\tilde{x}_3), \ldots, g^{-1}(\tilde{x}_n)\big) \\
&= -\sum_{g \in \mathcal{G}_\epsilon(x_k)} w_\epsilon^g(x_k) \, \psi_\theta\big(g^{-1}(\tilde{x}_1), g^{-1}(\tilde{x}_2), g^{-1}(\tilde{x}_3), \ldots, g^{-1}(\tilde{x}_n)\big) \\
&= -\psi_{\theta;\epsilon}^{(\mathrm{SC};k)}(\tilde{x}_1, \ldots, \tilde{x}_n) .
\end{aligned}$$

For the case $k = 1, 2$, we can apply a similar calculation while noting that the weights remain unchanged, and obtain

$$\begin{aligned}
\psi_{\theta;\epsilon}^{(\mathrm{SC};1)}(\tilde{x}_2, \tilde{x}_1, \tilde{x}_3 \ldots, \tilde{x}_n) &= -\psi_\theta^{(\mathrm{SC};2)}(\tilde{x}_1, \tilde{x}_2, \tilde{x}_3 \ldots, \tilde{x}_n) , \\
\psi_\theta^{(\mathrm{SC};2)}(\tilde{x}_2, \tilde{x}_1, \tilde{x}_3 \ldots, \tilde{x}_n) &= -\psi_{\theta;\epsilon}^{(\mathrm{SC};1)}(\tilde{x}_1, \tilde{x}_2, \tilde{x}_3 \ldots, \tilde{x}_n) .
\end{aligned}$$

This implies that

$$\psi_{\theta;\epsilon}^{(\mathrm{SC})}(\tilde{x}_{\pi(1)}, \ldots, \tilde{x}_{\pi(n)}) = -\frac{1}{n} \sum_{k=1}^{n} \psi_{\theta;\epsilon}^{(\mathrm{SC};k)}(\tilde{x}_1, \tilde{x}_2, \tilde{x}_3 \ldots, \tilde{x}_n) = \mathrm{sgn}(\pi) \, \psi_{\theta;\epsilon}^{(\mathrm{SC})}(\tilde{x}_1, \ldots, \tilde{x}_n) .$$

Since the choice of indices 1 and 2 are arbitrary, the above in fact holds for all transpositions $\pi$, which implies

$$\psi_{\theta;\epsilon}^{(\mathrm{SC})}(\tilde{x}_{\tau(1)}, \ldots, \tilde{x}_{\tau(n)}) = \mathrm{sgn}(\tau) \, \psi_{\theta;\epsilon}^{(\mathrm{SC})}(\tilde{x}_1, \ldots, \tilde{x}_n)$$

for all permutations $\tau$ of the $n$ electrons. This proves (ii).

To prove (iii), it suffices to show that for every $k \leq n$, the function $\psi_{\theta;\epsilon}^{(\mathrm{SC};k)}$ defined above is $p$-times continuously differentiable at $\mathbf{x}$, i.e. $\nabla^p \psi_{\theta;\epsilon}^{(\mathrm{SC};k)}$ exists and is continuous at $\mathbf{x}$. Again it suffices to show this for the case $k = 1$. Let $\tilde{\mathbf{x}} := (\tilde{x}_1, \ldots, \tilde{x}_n)$ be a vector in a sufficiently small neighborhood of a fixed $\mathbf{x} := (x_1, \ldots, x_n)$, and recall that

$$\mathcal{G}_\epsilon(\tilde{x}_1) = \big\{ g \in \mathbb{G} \,\big|\, d(\tilde{x}_1, g(\Pi_0)) \leq \epsilon \big\} .$$

For $\tilde{\mathbf{x}}$ in a sufficiently small neighborhood of $\mathbf{x}$, $\mathcal{G}_\epsilon(\tilde{x}_1)$ takes value in $\{\mathcal{G}_l\}_{0 \leq l \leq M}$, where

$$\mathcal{G}_l := \mathcal{G}_\epsilon(x_1) \setminus \{g_1^\epsilon, \ldots, g_l^\epsilon\} ,$$

and $g_1^\epsilon, \ldots, g_M^\epsilon \in \mathbb{G}$ is an enumeration of all group elements such that

$$d\big(x_1, g_l^\epsilon(\Pi_0)\big) = \epsilon \qquad \text{for } 0 \leq l \leq M .$$

Therefore $\psi_{\theta;\epsilon}^{(\mathrm{SC};1)}(\tilde{\mathbf{x}})$ takes values in $\{\psi_{\mathcal{G}_l}(\tilde{\mathbf{x}})\}_{0 \leq l \leq M}$, where

$$\psi_{\mathcal{G}_l}(\tilde{\mathbf{x}}) := \sum_{g \in \mathcal{G}_l} \frac{\lambda_\epsilon\big(\frac{\epsilon - d(\tilde{x}_1, g(\Pi_0))}{\epsilon}\big)}{\sum_{g' \in \mathcal{G}_l} \lambda_\epsilon\big(\frac{\epsilon - d(\tilde{x}_1, g'(\Pi_0))}{\epsilon}\big)} \, \psi_\theta\big(g^{-1}(\tilde{x}_1), \ldots, g^{-1}(\tilde{x}_n)\big) .$$

Notice that at $\tilde{\mathbf{x}} = \mathbf{x}$, by the definition of $g_l^\epsilon$, we have

$$\begin{aligned}
\psi_{\mathcal{G}_l}(\mathbf{x}) &= \sum_{g \in \mathcal{G}_l} \frac{\lambda_\epsilon\big(d(x_1, g(\Pi_0))\big)}{\sum_{g' \in \mathcal{G}_l} \lambda_\epsilon\big(d(x_1, g'(\Pi_0))\big)} \times \psi_\theta\big(g^{-1}(x_1), \ldots, g^{-1}(x_n)\big) \\
&\overset{(a)}{=} \sum_{g \in \mathcal{G}_\epsilon(x_1)} \frac{\lambda_\epsilon\big(d(x_1, g(\Pi_0))\big)}{\sum_{g' \in \mathcal{G}_\epsilon(x_1)} \lambda_\epsilon\big(d(x_1, g'(\Pi_0))\big)} \times \psi_\theta\big(g^{-1}(x_1), \ldots, g^{-1}(x_n)\big) = \psi_{\theta;\epsilon}^{(\mathrm{SC};1)}(\mathbf{x}) .
\end{aligned}$$

Since the above argument works with $\mathbf{x}$ replaced by $\tilde{\mathbf{x}}$, we also have

$$\psi_{\theta;\epsilon}^{(\text{SC};1)}(\tilde{\mathbf{x}}) = \sum_{g\in\tilde{\mathcal{G}}_l} \frac{\lambda_\epsilon\big(d(x_1, g(\Pi_0))\big)}{\sum_{g'\in\tilde{\mathcal{G}}_l} \lambda_\epsilon\big(d(x_1, g'(\Pi_0))\big)} \times \psi_\theta\big(g^{-1}(x_1), \ldots, g^{-1}(x_n)\big) \tag{12}$$

for $0 \le l \le \tilde{M}$, where we have defined

$$\tilde{\mathcal{G}}_l := \mathcal{G}_\epsilon(\tilde{x}_1) \setminus \{\tilde{g}_1^\epsilon, \ldots, \tilde{g}_l^\epsilon\},$$

and $\tilde{g}_1^\epsilon, \ldots, \tilde{g}_M^\epsilon \in \mathbb{G}$ is an enumeration of all group elements such that

$$d\big(\tilde{x}_1, \tilde{g}_l^\epsilon(\Pi_0)\big) = \epsilon \qquad \text{for } 0 \le l \le M.$$

Notice that for $\tilde{\mathbf{x}}$ in a sufficiently small neighborhood of $\mathbf{x}$, we have $\{\tilde{\mathcal{G}}_l\}_{l\le\tilde{M}} = \{\mathcal{G}_l\}_{l\le M}$, in which case (12) implies

$$\psi_{\theta;\epsilon}^{(\text{SC};1)}(\tilde{\mathbf{x}}) = \psi_{\mathcal{G}_l}(\tilde{\mathbf{x}}) \qquad \text{for all } 0 \le l \le M.$$

In other words, we have shown that in a sufficiently small neighborhood of $\mathbf{x}$, $F_1$ equals $\psi_{\mathcal{G}_l}$ for all $1 \le l \le M$. Recall that $\lambda$ and $d(\bullet, g(\Pi_0))$ are $p$-times continuously differentiable for all $g \in \mathbb{G}$, and $\psi_\theta$ is $p$-times continuously differentiable at $g(\mathbf{x})$ for all $g \in \mathbb{G}$ by assumption. This implies that $\psi_{\mathcal{G}_l}$ is also $p$-times continuously differentiable at $\mathbf{x}$ and so is $\psi_{\theta;\epsilon}^{(\text{SC};1)}$. Moreover, for $0 \le q \le p$, the derivative can be computed as

$$\nabla^q \psi_{\theta;\epsilon}^{(\text{SC};1)}(\mathbf{x}) = \nabla^q \psi_{\mathcal{G}_l}(\mathbf{x}) = \sum_{g\in\mathcal{G}_\epsilon(x_1)} \nabla^q \psi_{\theta;g,x_1}^{(\text{SC};k)}(\mathbf{x}),$$

where we recall that for $1 \le k \le n$, $g \in \mathbb{G}$ and $x, y_1, \ldots, y_n \in \mathbb{R}^3$,

$$\psi_{\theta,\epsilon;g,x}^{(\text{SC};k)}(y_1, \ldots, y_n) := \frac{\lambda_\epsilon\big(d(y_k, g(\Pi_0))\big)}{\sum_{g'\in\mathcal{G}_\epsilon(x)} \lambda_\epsilon\big(d(y_k, g'(\Pi_0))\big)} \psi_\theta\big(g^{-1}(y_1), \ldots, g^{-1}(y_n)\big).$$

The same argument applies to all $\psi_{\theta;\epsilon}^{(\text{SC};k)}$'s with $k \le n$ and therefore for $0 \le q \le p$,

$$\nabla^q \psi_{\theta;\epsilon}^{(\text{SC})}(x_1, \ldots, x_n) = \frac{1}{n} \sum_{k=1}^n \nabla^q \psi_{\theta;\epsilon}^{(\text{SC};k)}(x_1, \ldots, x_n) = \frac{1}{n} \sum_{k=1}^n \sum_{g\in\mathcal{G}_\epsilon(x_k)} \nabla^q \psi_{\theta,\epsilon;g,x_k}^{(\text{SC};k)}(\mathbf{x}),$$

which proves (iii). $\qquad\square$

## H.2. Proof of Lemma E.2

Since $\lambda_\epsilon(0) = 1$, $\lambda_\epsilon(\epsilon) = 0$ and $\lambda_\epsilon$ is continuously differentiable, by the mean value theorem, there is $y_1 \in (0, \epsilon)$ such that

$$\partial\lambda_\epsilon(y_1) = (1-0)/\epsilon = \epsilon^{-1}.$$

Meanwhile, since $\lambda_\epsilon(w) = 1$ for all $w \le 0$, $\partial\lambda_\epsilon(w) = 0$ for all $w < 0$, and since $\partial\lambda_\epsilon$ is continuous, $\partial\lambda_\epsilon(0) = 0$. By the twice continuous differentiability of $\lambda_\epsilon$ and the mean value theorem again, there exists $y_1' \in (0, y_1) \subset (0, \epsilon)$ such that

$$\partial^2\lambda_\epsilon(y_1') = (\epsilon^{-1} - 0)/y_1 = \frac{1}{\epsilon y_1'} \ge \frac{1}{\epsilon^2}.$$

Since $\partial^2\lambda_\epsilon(w) = 0$ for all $w < 0$ and $\partial^2\lambda_\epsilon$ is continuous, $\partial^2\lambda_\epsilon(0) = 0$. As $\partial^2\lambda_\epsilon(y_1') \ge \frac{1}{\epsilon^2}$, by the intermediate value theorem, there exists some $y_2 \in [0, y_1] \subseteq [0, \epsilon]$ such that $\partial^2\lambda_\epsilon(y_2) = \epsilon^{-2}$. $\qquad\square$

## I. Discussion on the case where per-iter sampling cost is greater than per-iter gradient cost

An anonymous reviewer has pointed out regimes where, unlike Section 3, $C_{\text{samp}}$ may take larger values than $C_{\text{grad}}$. We briefly discuss this point. There are two cases where $C_{\text{samp}} \gg C_{\text{grad}}$ may occur: (i) There exists significantly accelerated implementations of gradient computation (Li et al., 2024) such that it becomes more efficient than sampling; (ii) One typically needs to scale the number of sampling steps and neural network size both with the physical system size (von Glehn et al., 2023; Li et al., 2022), and the increase in per-step sampling cost could outweigh the increase in per-step gradient cost. In these regimes, our theoretical claims do hold, but come with some additional nuances:

- **DA instability.** When $C_\mathrm{samp} \ll C_\mathrm{grad}$, our DA batch size in Section 4.1 was chosen as $k \times N' = N$, where $N' := N/k$ is the number of i.i.d. drawn samples and $k$ is the number of augmentations. The reduction in the number of i.i.d. drawn samples was key to the destabilization effect. If instead $C_\mathrm{samp} \gg C_\mathrm{grad}$, one may increase $N'$, the number of i.i.d. samples drawn, though the same conclusion holds as long as $N' < N$. Meanwhile if $N' = N$, DA does not destabilize, but the $k$ augmentations always increase the computational cost. The statistical-computational tradeoff now depends on how large $N'$ is and therefore has a more intricate dependence on the computational cost ratio $C_\mathrm{samp}/C_\mathrm{grad}$.

- **GA instability.** The destabilization effect still occurs, as group-averaging always increases both sampling and gradient evaluation cost, which necessitates a reduction in the number of i.i.d. samples drawn.

- **PA benefits.** PA is still computationally attractive compared to in-training symmetrizations, since the overall training cost still typically outweighs the inference cost.

Empirically, we do not observe $C_\mathrm{samp} \gg C_\mathrm{grad}$ for the DeepSolid experiments we have performed.

