# OpenReview forum: "Diagonal Symmetrization of Neural Network Solvers for the Many-Electron Schrödinger Equation"
_ICML.cc/2025/Conference — ICML 2025 poster_

### Official Review · Reviewer_vG6b · 2025-03-10

**Overall Recommendation:** 2

**Summary:**

This work investigates the impact of symmetrization in neural network wave functions for periodic solid systems. Specifically, the authors compare data augmentation, group averaging, and canonicalization. Contrary to other fields, the authors find that symmetrization may hurt performance, but post-hoc averaging generally yields better estimates.

## After the rebuttal

I increased my score in light of the authors’ rebuttal. However, I remain skeptical about the correctness of the evaluation. Biased estimates in VMC have frequently been a point in NN-VMC, and the reason why the number of MCMC steps typically increases linearly with the system size, as seen in [1]. Still, my main concern is the limited scope of this work. While it is certainly very valuable, I find the current scope too limited for a top-tier conference. I encourage the authors to broaden their evaluation, supporting wider applicable statements.

[1] von Glehn et al. "A Self-Attention Ansatz for Ab-initio Quantum Chemistry"

**Claims And Evidence:**

The authors make the following claims:
1. symmetrization to diagonal groups destabilizes training and can lead to worse performance.
2. post hoc averaging is effective in improving neural network solvers.

The paper supports these claims but to a somewhat limited degree due to the limited evaluation and specific problem statement. There appear to be theoretical misunderstandings about how to compute energy gradients in VMC. This most likely invalidates the data augmentation results. In general, contrary to the text statement, computing energy gradients does not involve computing $\partial_\theta\partial^2_x\psi_\theta$ but is done by taking the derivative of $\nabla_\theta E_{\psi^2}[\psi^{-1}H\psi]$ which corresponds to $E[(E_L - E[E_L])\nabla_\theta\ln\psi]$ where $E_L=\psi^{-1}H\psi$ as the Hamiltonian is hermitian, i.e., $E[\nabla_\theta E_L]=0$. Thus, the gradient we're computing is $\int (\nabla \frac{\psi(x)^2}{\langle\psi\vert \psi\rangle}) E_L(x) dx$. Getting the expectations right is important. In the data averaging setup, the distribution of the expectations is changed yielding incorrect estimates and gradients. Further, contrary to standard data augmentation, a batch always includes the original sample and all augmentations, which is not the case in the standard case.

Another problem to support these claims is that the evaluation is limited to a single neural network wave function, limiting the impact of the statements on this neural network and the concrete symmetry groups. However, I acknowledge that the list of known symmetries is short for open boundary conditions.

**Essential References Not Discussed:**

The authors could discuss the difference to canonicalization schemes like [1,2] which focus on obtaining the right symmetries for observables rather than the wave function itself.

[1] Gao et al. "Ab-Initio Potential Energy Surfaces by Pairing GNNs with Neural Wave Functions"
[2] Gerard et al. "Gold-standard solutions to the Schrödinger equation using deep learning: How much physics do we need?"

**Experimental Designs Or Analyses:**

See above regarding the validity of the data augmentation scheme. Additional wave functions would be interesting, e.g., [1].

[1] Gerard et al. "Transferable Neural Wavefunctions for Solids"

**Methods And Evaluation Criteria:**

The chosen periodic compounds are well chosen but the selection could be expanded to a cover a larger number of systems.

**Other Comments Or Suggestions:**

The justification for not including DA as a post hoc method is very obscure (l. 358), but given that it would yield incorrect estimates, it should anyway not be included.
The statement in robustnuss to outliers about the laplacian is incorrect; in practice, one does not compute $H\psi$ and then divides by $\psi$ but instead directly uses the formulation $\psi^{-1} H \psi=\nabla^2\ln\psi + (\nabla\ln\psi)^2$.

**Other Strengths And Weaknesses:**

The authors strongly focus on the negative results. Unfortunately, I don't see these supported very well due to the limited evaluation and partially flawed evaluation. The post-hoc averaging technique presents a valuable improvement for future solid VMC calculations but may be too limited in scope to justify an acceptance. Extending the scope to other structures and models or finding efficient symmetrization techniques that scale well would strengthen the position of the paper.

**Questions For Authors:**

* Is there a way to translate these insights to open boundary conditions?
* Can we incorporate the right symmetries directly in the architectures like SO(3)-equivariant force fields?

**Relation To Broader Scientific Literature:**

Prior literature in neural network VMC focused primarily on accurate calculations by opening all degrees of freedom. This is mostly because most works focus on open boundary conditions where little is known about the symmetries of the wave function itself, only about the symmetries of its observables.

**Theoretical Claims:**

It is unclear to me what the message of Proposition 4.1 is. I generally found the notation unnecessarily complicated and partly undefined, e.g., the definitions of F and Q in l. 169/170.

Whether sampling or gradient computations dominate the computations typically depends on the system size in VMC. For many electrons, sampling may take more time; e.g., in [1], the authors increase the number of sampling steps with the system size, and the gradient computation can be significantly accelerated [2].

[1] von Glehn et al. "A Self-Attention Ansatz for Ab-initio Quantum Chemistry"
[2] Li et al. "A computational framework for neural network-based variational Monte Carlo with Forward Laplacian"

---

> ### Author Rebuttal · Authors · 2025-03-30
>
> We thank the reviewer for the constructive feedback and apologize for any confusion that arise from presentation issues or differences in understanding, which we now address.
>
> ---
> **Correctness of gradient.** We thank the reviewer for the typo: $\partial_\theta \partial_x^2$ should indeed be $\partial_x^2$. The expectation is **otherwise correct**: F and Q are just generic update terms that
> - accommodate specific implementations of the gradient e.g. KFAC;
> - allow Prop 4.1, Lem 4.2, Thm D.1 to be *not tied to* a specific implementation.
>
> We also stress that **the reviewer's formula is exactly the one that we compute numerically**; DA is run directly on the original DeepSolid except that gradient is computed on an augmented set of samples.
>
> ---
> **Validity of DA,** which seems to be the main concern.
> - We stress that DA is *not* the method we champion. It's a popular ML method that one expects to help under approx. invariance; we follow its standard usage and discuss how it can fail.
> - As stated in Sec. 4.1, we follow standard usage to randomly sample DA **with replacement** (note the i.i.d. part). Unlike what the review suggests, the data is **not guaranteed** to include *either* the original *or* all the augmented data.
> - Indeed DA computes $E[F_{X,\psi}]$ under a different distribution of X without changing $\psi$. Yet this gives a **biased estimate, not an invalid estimate**. There are many other sources of bias e.g. stochastic gradient, KFAC approx etc., **some of which are known to help training**, and the gradient estimate is never unbiased in practice. The DA estimate is physically valid when $\psi$ is exactly invariant -- where DA does nothing -- and gives a regularized estimate when $\psi$ is approx. invariant. In ML, DA is routinely used under approx. invariance ([6,7,8] cited in the response to Reviewer mfuU) and **its bias is known to enforce regularization towards symmetry** ([5,9] cited in the response to Reviewer mfuU). For us, approx. invariance of $\psi$ is enforced by pre-training and visible in Fig. 5(c). Moreover, DA gives **a physically valid energy** (Fig. 5(a)), just not an improved one. **We do agree this is important to clarify and will add this point.**
> ---
> **Sampling vs gradient cost.** We thank the reviewer for highlighting the nuance that there may be better algorithms s.t. per-iter gradient compute is faster than per-iter sampling, esp. if the system requires more MCMC steps for good sampling. While we don't observe this in our setup, **we'll address it in the discussion**. We stress that the per-iter compute comparison is *not central* to our arguments, with some small additional nuances:
> - DA instability: Our DA batch size is k x N/k = N. If sampling dominates, one may exploit the sampling speedup by DA to get N' > N/k samples. Yet if N' < N, the use of fewer i.i.d. samples still inflates the variance; if N'=N, DA doesn't destabilize but the k DAs always increase compute cost. The tradeoff now depends on how large N' can be and thus how fast gradient compute is v.s. sampling.
> - GA: Unchanged; both sampling and gradient costs increase.
> - PA: Still computationally attractive, as overall training cost typically outweighs inference cost. The per-iter MCMC steps is multiplied by a large no. of training iters.
> ---
>
> **Correctness of robustness.** We stress that the reviewer gives a practically useful but **mathematically equivalent** formula to $\psi^{-1} H \psi$, to which the *exact same discussion* is true, just more cumbersome. Note that differentiating $\ln \psi$ gives $\psi^{-1}$.
>
> ---
> **Over-emphasis on the negative results; more scalable symmetrization needed.** We note that our work **mainly seeks to offer perspectives rather than a universally good symmetrization**: We study a particularly hard case of symmetries and examine the strenghts and limitations of known ML symmetrizations. Given the success of posthoc averaging, one naturally expects in-training averaging / augmentations would help even more. **Our negative results are crucial in examining their potential failure points** and why that's not the case for PA. We also expect the negative results to be of **independent interest** to the applications of DA and GA **beyond a physics context**. We do agree that finding more scalable symmetrization for a large class of setups is an interesting open problem. Yet as the reviewer suggested, this requires finding the best combination of architecture, optimizer and symmetrization, and it's hard to tell whether the improvement is from symmetry or from other factors; see response 1.3) to szMY. We believe **part of the merits of our work is on an apples-to-apples comparison on a fixed architecture with v.s. without symmetry.** We will clarify these in the revision.
>
> ---
> **References and questions**: See response 1) to szMY.
>
> ---
> If the above help to answer essential concerns behind the rejection, e.g. validity of DA, we'd be grateful if you'd consider raising the score.

---

### Official Review · Reviewer_wZs6 · 2025-03-14

**Overall Recommendation:** 4

**Summary:**

The paper studies diagonal group symmetries in neural network solvers for many-electron Schrödinger equations, comparing different symmetrization approaches: data augmentation (DA), group averaging (GA), and post-hoc averaging (PA). The main claim is that in-training symmetrization can hurt performance while post-hoc averaging helps.

**Claims And Evidence:**

The paper's theoretical analysis of gradient variance (Proposition 4.1) is well-derived and shows how DA introduces a variance inflation factor (k-1)/N. This manifests empirically in Figure 3, where DA with k=12 shows ~1.5x higher normalized gradient variance compared to baseline.

The post-hoc averaging results are compelling for the tested systems. e.g. improves energy from -8.138 to -8.1507 and reduces variance. However, the paper only tests three systems all using the DeepSolid architecture. Testing on more architectures (e.g., FermiNet, PauliNet) would strengthen the claims about PA's general effectiveness.

The claim that "in-training symmetrization destabilizes training" seems to conflict with the finding in (https://journals.aps.org/prresearch/abstract/10.1103/PhysRevResearch.5.013216), which demonstrates that properly incorporating symmetries into network architecture can actually improve training stability and efficiency in quantum lattice models. While the settings are different (lattice vs continuous systems), this contradiction deserves more discussion. The authors' theoretical analysis focuses on computational-statistical tradeoffs, but may be missing other mechanisms through which symmetries could aid optimization.

**Essential References Not Discussed:**

https://www.nature.com/articles/s41567-024-02566-1 (second order optimization)

https://journals.aps.org/prresearch/abstract/10.1103/PhysRevResearch.5.013216 (symmetry)

And many other foundational works in VMC are also missing, including
https://www.science.org/doi/10.1126/science.aag2302
and many other works from the authors of the papers mentioned above

**Experimental Designs Or Analyses:**

The training setup uses KFAC optimization with batch sizes adjusted for computational budget. However, the sensitivity to these choices isn't explored - would different batch size ratios change the relative performance?

The MCMC chain length of 30000 steps for evaluation seems reasonable but isn't justified. Given that PA's benefits come from better sampling, analyzing how results vary with chain length would be valuable.

**Methods And Evaluation Criteria:**

The evaluation uses standard VMC metrics: ground state energy and local energy variance. The energy improvements are significant given that chemical accuracy is ~0.00159. However, the paper lacks evaluation of other physical properties like electron density or correlation functions that could provide additional validation.

The computational cost analysis in Table 1 is detailed but raises questions. For graphene 1x1, GA with N=1000, k=12 takes 25s per step versus 2.5s for baseline, which is a 10x slowdown. The paper doesn't fully explain this large difference given that only k=12 group operations are being averaged. Additionally, the choice of KFAC optimizer is not justified - recent work has shown other second-order methods can be more effective for quantum many-body problems (e.g. https://www.nature.com/articles/s41567-024-02566-1).

**Other Comments Or Suggestions:**

No other comments.

**Other Strengths And Weaknesses:**

Already mentioned above.

**Questions For Authors:**

1. For LiH, PA with F222 (k=16) achieves most of the improvement of Fm$\bar{3}$m (k=192). Is there a systematic way to choose minimal symmetry groups that capture most benefits?

2. The gradient variance analysis assumes first-order optimization, but experiments use KFAC. Can the theory be extended to second-order methods? Would this explain why GA performs better than DA despite similar variance bounds?

3. For bcc-Li, different subgroups (P4/mmm, Fmmm) give similar improvements despite capturing different symmetries. What determines which symmetries are most important for improving the wavefunction?

4. Why was KFAC chosen as the optimizer? Have you compared with other second-order methods like (https://www.nature.com/articles/s41567-024-02566-1)

5. How does your post-hoc averaging approach compare with other symmetry preservation methods (e.g. https://journals.aps.org/prresearch/abstract/10.1103/PhysRevResearch.5.013216, where they show incorporating symmetries architecturally can improve training in quantum lattice models.

6. How does your post-hoc averaging approach handle solutions allowing for different symmetry sectors?

**Relation To Broader Scientific Literature:**

The paper provides some coverage of recent neural network VMC methods but comparisons with other symmetry approaches are limited.

**Theoretical Claims:**

The high-dimensional CLT result (Theorem D.1) provides rigorous bounds on the distribution of gradient updates. However, the key assumption that seems strong and its practical validity isn't verified empirically.

The smoothed canonicalization analysis shows an O(nk) cost scaling with number of electrons and group size. This explains why SC performs poorly in practice, but the paper could better explain if this limitation is fundamental or implementation-specific.

---

> ### Author Rebuttal · Authors · 2025-03-30
>
> We thank the reviewer for the constructive feedback, which we address below.  We will add **a new discussion and limitation section** to reflect the discussions.
>
> ---
> 1) **PA's effectiveness on more architectures.** Indeed it's interesting whether PA is effective beyond DeepSolid. We stress that, as motivated in Sec. 2, we focus on a class of difficult restricted symmetries, i.e. diagonal space groups, for which only **model-agnostic symmetrizations are known**, which lead to our choice of DA, GA and PA. In the context of these symmetries, **DeepSolid is the only architecture we know that performs well on infinite periodic solids**, and is itself based on FermiNet. We do expect that for the *non-solid* systems handled by the original FermiNet and PauliNet, **architectural symmetrization** (e.g. the invariant maps discussed in our Sect. 2) may be possible and can be more effective than PA -- see response 1) to szMY.
> ---
> 2) **Conflicts with known benefits of symmetries.** The seemingly conflicting conclusions is related to point 1 in that we were forced to consider *model-agnostic symmetrizations*. For symmetries where *architectural symmetrization* is known, we don't expect them to suffer from the same cost tradeoffs or the same instability we saw for DA and GA. **We agree that symmetries may act via mechanisms we didn't address.** Here, we **try to minimize the influence of other mechanisms** in 2 ways: a) Focusing on the apples-to-apples comparison on a **fixed architecture with fixed optimizer and hyperparameters** and only vary the symmetrization; b) Citing a high-d CLT result that reduces the effect of DA to bias and variance (considered in a rich body of work on Gaussian universality and e.g. [5] cited in the response to Reviewer mfuU), and understand their effects empirically by stability and energy performance. **There could be other mechanisms that we didn't observe, and we'll mention this as a limitation.** We also stress that, in view of recent findings in AI4Science (see our response 1.3) to szMY), whether symmetry helps with performance is an open question in general, and **our apples-to-apples comparison consitutes a concrete attempt to minimize confounding effects** in the cases of DA, GA and PA.
> ---
> 3) **10x compute difference.** Apologies if it's unclear. GA with 12 symmetries requires 12x more evaluations of the network at the different inputs, which is expected to increase the cost significantly. This is why batch size was set to N/k to keep the same cost.
> ---
> 4) **KFAC vs alternatives.** We choose KFAC as it's the choice for DeepSolid and to ensure a fair comparison with the original model; see 2) above.
> ---
> 5) **Explanations of SC.** The limitation is specific to the ``average-near-the-boundary" approach of building SC. Detailed explanations are in Appendix E, as the precise mathematical setup of SC is tedious (also the case in Dym et al. '24, who proposed the SC method we adapt). **Whether an efficient SC is possible for complicated groups is an open theoretical question:** Dym et al. only recently establish impossibility results for specific groups, and as mentioned in our Sec 2, building SC for diagonal space group is related to the problem of maximal invariants and orbifolds, which are unsolved math problems. **We'll ensure these are clarified in the SC section.**
> ---
> 6) **Batch-size ratios.** If we read the qn correctly, different ratios **are already investigated** in Fig. 7 of Appendix B.4.
> ---
> 7) **MCMC length.** We picked 30k as we empirically found that the sampled values had stabilised at this length. To address the review, we obtained additional runs for LiH PA with F222: The energy and var values are (-8.147(1), 0.018(1)) for 20k and (-8.1486(9), 0.0167(9)) for 40k, both within error range of the 30k result. Note that PA's benefits aren't from better quality of sampling but from the fact that it is sampling a different wavefunction.
> ---
> 8) **References; compare with other symmetrizations.** See above comments and response 1 to szMY.
> ---
> 9) **Theory for 2nd order methods.** This was discussed at the end of Sec 4.1, which highlights that the key barrier is analysing the high-dimensional Hessian matrix. As a further remark, a precise analysis rests on analysing the eigenvalues of a large random matrix; when the matrix does not have i.i.d. entries, this is a known hard problem in random matrix theory and no general results are yet known.
> ---
> 10) **Choice of groups.** The interesting observations in LiH and bcc-Li are indeed why we include the results. We don't have a concrete answer, but conjecture the effects to be a) system-specific and b) dependent on what approx. invariances the wavefunction was pretrained to possess. **We'll discuss these as interesting directions for future work.**
> ---
> We hope we've addressed as many comments as possible within the response limit. If they are helpful in addressing your concerns, we would be grateful if you would consider raising the score.

---

> > ### Comment · Reviewer_wZs6 · 2025-04-04
> >
> > Thanks for the detailed rebuttal. I have raised my score.

---

### Official Review · Reviewer_szMY · 2025-03-17

**Overall Recommendation:** 3

**Summary:**

This paper investigated different methods for incorporating diagonal symmetries in neural network solvers for the many-electron Schrödinger equation, with a particular focus on variational Monte Carlo (VMC) methods. Specifically, the authors studied three main approaches to enforce diagonal invariance: data augmentation, group averaging and canonicalization (adding invariant features). Surprisingly, the authors found that in-training symmetrization often hurts performance, while posthoc averaging improves accuracy and symmetry of the learned wavefunctions. Both theoretical and empirical results are presented to justify the efficacy of posthoc averaging.

## Update After Rebuttal

The reviewer is generally positive about the results presented in the paper and satisfied with the authors' responses to the questions raised here. Hence, the reviewer would like to remain the score.

**Claims And Evidence:**

Here is the main claim made in this paper: posthoc averaging is a better choice to enforce diagonal invariance compared to in-training symmetrization. Sufficiently many experiments on Graphene, Lithium Hydride (LiH) and Metallic Lithium (bcc-Li) are included to compare postdoc averaging with other in-training symmetrization methods like data augmentation and group averaging, which supports the main claim in a clear and convincing way.

**Essential References Not Discussed:**

The reviewer finds that several key references are not discussed in this work. For instance, the authors didn't cite [1], which is, to the best of the reviewer's knowledge, one of the pioneering works that try to solve the many-body Schrödinger equation based on neural networks. Furthermore, in addition to the literature cited in the article, some other work like [2,3,4,5,6,7] have also studied the problem of incorporating symmetry within the machine learning based solvers of many-body Schrödinger equation. It might be meaningful for the authors to cite these articles and briefly discuss them as related work.

References:

[1] Carleo, Giuseppe, and Matthias Troyer. "Solving the quantum many-body problem with artificial neural networks." Science 355, no. 6325 (2017): 602-606.

[2] Mahajan, Ankit, and Sandeep Sharma. "Symmetry-projected Jastrow mean-field wave function in variational Monte Carlo." The Journal of Physical Chemistry A 123, no. 17 (2019): 3911-3921.

[3] Han, Jiequn, Linfeng Zhang, and E. Weinan. "Solving many-electron Schrödinger equation using deep neural networks." Journal of Computational Physics 399 (2019): 108929.

[4] Zepeda-Núñez, Leonardo, Yixiao Chen, Jiefu Zhang, Weile Jia, Linfeng Zhang, and Lin Lin. "Deep Density: circumventing the Kohn-Sham equations via symmetry preserving neural networks." Journal of Computational Physics 443 (2021): 110523.

[5] Lin, Jeffmin, Gil Goldshlager, and Lin Lin. "Explicitly antisymmetrized neural network layers for variational Monte Carlo simulation." Journal of Computational Physics 474 (2023): 111765.

[6] Abrahamsen, Nilin, Zhiyan Ding, Gil Goldshlager, and Lin Lin. "Convergence of variational Monte Carlo simulation and scale-invariant pre-training." Journal of Computational Physics 513 (2024): 113140.

[7] Zhang, Yaolong, Bin Jiang, and Hua Guo. "SchrödingerNet: A Universal Neural Network Solver for the Schrödinger Equation." Journal of Chemical Theory and Computation 21, no. 2 (2025): 670-677.

**Experimental Designs Or Analyses:**

Yes, please refer to the "Methods And Evaluation Criteria" section above.

**Methods And Evaluation Criteria:**

Yes. The experiments are mainly based on Graphene, Lithium Hydride (LiH) and Metallic Lithium (bcc-Li), which are standard examples used in the DeepSolid paper. The authors studies different symmetrization methods (data augmentation, group averaging and posthoc averaging) and compare their performance based on the induced wave functions and associated quantities like the ground state energy.

**Other Comments Or Suggestions:**

N/A

**Other Strengths And Weaknesses:**

This paper used a mixture of theory and practical experiments to justify the advantage of posthoc averaging compared to other approaches like data augmentation and group averaging, which is one of the first work that compares different ways of enforcing symmetry in the context of solving high-dimensional many-body Schrödinger equation. However, some potential drawbacks also exist. Firstly, the study is mainly limited to the Variational Monte Carlo (VMC) solver, so it might be meaningful to explore other solvers like Diffusion Monte Carlo (DMC) as well - see for instance [1]. Secondly, it seems that the claims and proofs in the paper can be made in a more mathematically rigorous way (which can be left as future work). For instance, below are some related literature that studies solving high-dimensional partial differential equations (PDEs) and symmetry from the perspective of statistical learning theory: [2,3,4,5,6,7,8,9,10].

References:

[1] Han, Jiequn, Jianfeng Lu, and Mo Zhou. "Solving high-dimensional eigenvalue problems using deep neural networks: A diffusion Monte Carlo like approach." Journal of Computational Physics 423 (2020): 109792.

[2] Jiao, Yuling, Yanming Lai, Dingwei Li, Xiliang Lu, Fengru Wang, Yang Wang, and Jerry Zhijian Yang. "A rate of convergence of physics informed neural networks for the linear second order elliptic pdes." arXiv preprint arXiv:2109.01780 (2021).

[3] Duan, Chenguang, Yuling Jiao, Yanming Lai, Xiliang Lu, and Zhijian Yang. "Convergence rate analysis for deep ritz method." arXiv preprint arXiv:2103.13330 (2021).

[4] Lu, Yiping, Haoxuan Chen, Jianfeng Lu, Lexing Ying, and Jose Blanchet. "Machine learning for elliptic PDEs: Fast rate generalization bound, neural scaling law and minimax optimality." arXiv preprint arXiv:2110.06897 (2021).

[5] Lu, Jianfeng, and Yulong Lu. "A priori generalization error analysis of two-layer neural networks for solving high dimensional Schrödinger eigenvalue problems." Communications of the American Mathematical Society 2, no. 1 (2022): 1-21.

[6] Lu, Yulong, Jianfeng Lu, and Min Wang. "A priori generalization analysis of the deep Ritz method for solving high dimensional elliptic partial differential equations." In Conference on learning theory, pp. 3196-3241. PMLR, 2021.

[5] Zweig, Aaron, and Joan Bruna. "Symmetric single index learning." arXiv preprint arXiv:2310.02117 (2023).

[6] Zweig, Aaron, and Joan Bruna. "Exponential separations in symmetric neural networks." Advances in Neural Information Processing Systems 35 (2022): 33134-33145.

[7] Zweig, Aaron, and Joan Bruna. "Towards antisymmetric neural ansatz separation." arXiv preprint arXiv:2208.03264 (2022).

[8] Zweig, Aaron, and Joan Bruna. "A functional perspective on learning symmetric functions with neural networks." In International Conference on Machine Learning, pp. 13023-13032. PMLR, 2021.

[9] Soleymani, Ashkan, Behrooz Tahmasebi, Stefanie Jegelka, and Patrick Jaillet. "Learning with Exact Invariances in Polynomial Time." arXiv preprint arXiv:2502.19758 (2025).

[10] Tahmasebi, Behrooz, and Stefanie Jegelka. "The exact sample complexity gain from invariances for kernel regression." Advances in Neural Information Processing Systems 36 (2023).

**Questions For Authors:**

N/A

**Relation To Broader Scientific Literature:**

This work is mainly posited within the AI4Science literature on solving high-dimensional many-body Schrödinger equation based on wavefunctions parametrized via neural networks, which has applications in physics, chemistry and material sciences. In terms of methodology used in this paper, it mainly falls within the category of symmetry + machine learning.

**Theoretical Claims:**

The only theoretical claim in the main text of the paper is Proposition 4.1. Its proof in Appendix G has been verified to be correct.

---

> ### Author Rebuttal · Authors · 2025-03-30
>
> We thank the reviewer for the constructive feedback. As all reviewers suggest additional references with some overlaps and similar questions on extendability, and due to the 5000 character limit per response, we address them together below.
>
> ---
> 1) **Essential references.** We thank all reviewers for mentioning interesting works. In the revision, we will **extend the introduction** and **include a new discussion and limitation section** to discuss these works. In detail,
>
>     **1.1) Essential pioneering works on neural network VMC solvers:** We will highlight [1] suggested by szMY (and also by wZs6) and other works by authors of [1].
>
>     **1.2) Comparisons to symmetrizations that do not address diagonal space group symmetries or architectures that are not known to be effective in solids.** We first stress that we focus on **a class of difficult restricted symmetries**, i.e. diagonal space groups, which notably has only *discrete* point group symmetries. As observed by Reviewer wZs6, this is quite different from continuous symmetries like SO(2) and SU(2). **We realize this may not be sufficiently emphasized in our abstract and intro and will do so in the revision.** Also see our Section 2 for why these restricted symmetries are harder than e.g. the richer continuous symmetries in E(3), where the same discussion can be extended to SO(2) and SU(2). Architecture-wise, we stress that in the context of diagonal space group symmetries, **DeepSolid is the only architecture we know that performs well on infinite periodic solids**. Reviewer vG6b has pointed out interesting papers under Essential References and notes alternative approaches that focus on symmetries of observables rather than of wavefunctions. We agree that these are parallel and complementary approaches of interest and will cite these works. We clarify that for solids, our considered symmetries are typically applicable to ground state wavefunctions that are described by states with zero crystal momentums. We also note that, it is still an open question whether the architectures in those papers are effective in infinite period solids (since no solid benchmarks are reported) or for addressing the symmetries of the periodic solid. Making them work with solids would require additional tweaks, and **it will not be clear whether performance improvements or drops come from symmetrization or from additional architecture / hyperparameter tweaks**. This would invalidate our apples-to-apples comparison. However, we **wholeheartedly agree that it is important to survey more existing works on other symmetries**. We will cite [2,3,4,5,7] suggested by szMY and the symmetry papers suggested by wZs6 and vG6b, and highlight the differences. See also the response to wZs6 about why our findings seemingly contradict known benefits of symmetries.
>
>     **1.3) Extensions to broader contexts.** **We agree with reviewer szMY** that it is interesting to understand the effects of DA, GA and PA in other solvers like DMC. **We also agree with reviewer vG6b that** it is interesting whether our insights are applicable to open boundary problems and SO(3)-equivariant force fields. We do want to stress that **our findings are specific to complicated symmetries for which natural invariant maps are unknown**, which force us to adopt *model-agnostic approaches from conventional ML*. For simple symmetries or continuous symmetries like SO(3), where many *architectural symmetrization approaches* are available, we expect different results, e.g. no tradeoffs in in-training symmetrization. Nevertheless, **given the recent findings in protein structures and atomic potential ([1] and [2] cited in the response to Reviewer mfuU) that symmetries are unnecessary for training**, it is indeed of general interest how much symmetry plays a role in performance versus other factors e.g. hyperparameters. Our work can be viewed as **a first step towards this question in VMC**, by doing the first apples-to-apples comparison on an architecture before and after symmetrization. **The suggested extensions by the reviewers are very interesting future steps and we will mention them in the new discussion and limitation section.**
> ---
> 2) **Theory papers, szMY.** We thank the reviewer for suggesting these very interesting works on analysing high-dimensional Schrödinger equations (including [6] suggested by the reviewer in **essential references**). Indeed, our analyses are constrained by our desire to stay as close as possible to the DeepSolid VMC setup; a more careful analysis involves overcoming theoretical obstacles such as convergence guarantee of KFAC and short MCMC chains in high dimensions. We will discuss these suggestions as future directions to explore.
> ---
> We hope these answer your questions. If our responses are helpful, we would be grateful if you would consider raising your score.

---

> > ### Comment · Reviewer_szMY · 2025-04-06
> >
> > The reviewer would like to thank the authors for the detailed response, which has addressed most questions raised in the reviews. Overall, the reviewer is positive about the results and would like to remain the score.

---

### Official Review · Reviewer_mfuU · 2025-03-28

**Overall Recommendation:** 4

**Summary:**

This paper investigates methods for incorporating diagonal group symmetries into neural network wave function ansatze for solving the many-electron Schrödinger equation via Variational Monte Carlo (VMC). The authors compare three main approaches: data augmentation (DA), group averaging (GA) and canonicalization. The central claim is that, unlike typical machine learning scenarios, explicitly enforcing symmetry in-training can destabilize the optimization process and potentially lead to worse performance compared to post-hoc symmetrization. The methods are evaluated using the DeepSolid architecture on systems like H2, LiH, and graphene. The paper also introduces a method for visualizing diagonal symmetries in the high-dimensional wavefunction space.

## Update after rebuttal

Thank you for the detailed rebuttal. Your clarifications on the broader context has convinced me about it's relevance to ICML. I am increasing my score accordingly and look forward to the revised manuscript.

**Claims And Evidence:**

The central claims are: in-training symmetrization (specifically group averaging) can destabilize VMC training and lead to poorer performance; post hoc averaging (PA) improves energy, variance, and symmetry effectively and efficiently. The claims are supported by both proofs and empirical results across different crystalline solids.

**Essential References Not Discussed:**

I cant definitively identify essential missing references within the core VMC or equivariant ML literature for diagonal groups. The paper appears to cite key works in neural network wavefunctions, general ML symmetrization and VMC.

**Experimental Designs Or Analyses:**

As stated in methods and evaluation criteria, the experimental design and analyses are reasonable, with good baselines, datasets, metrics and ablation studies.

**Methods And Evaluation Criteria:**

Yes, the methods and evaluation criteria are largely appropriate for the problem of developing symmetry-aware neural network solvers for quantum many-body problems.

* The paper investigates symmetrization techniques: DA, GA, PA and PC
* The authors use appropriate metrics (energy and variance in in local energy) and propose a new metric Var[PA/OG] to provide a quantitative measure for symmetry. They also use GPU hours as a measure of computational cost.
* The models are benchmarked on 3 crystalline solids with DeepSolid as the baseline.

**Other Comments Or Suggestions:**

1. The paper could benefit from a brief discussion of how the findings might apply to other areas of physics simulation where similar symmetry constraints arise.
2. The visualization method in App C is a nice addition.

**Other Strengths And Weaknesses:**

The paper makes useful claims about properties of different symmetriziation approaches, including the counterintutive claim (from a standard ML viewpoint), that data augmentation can destablilize training. It provides a systematic comparison of these approaches applied during training and post-hoc. It is well-written and supports its main claims with both proofs and emperical results.

While this is a well-written paper with detailed proofs and emperical results, I am concerned that it is too specialized for a venue like ICML and might be more suited to a physics journal.

**Questions For Authors:**

How sensitive are the results, particularly the ranking of the methods (OG, DA, GA, GAs, PA), to the choice of neural network architecture and hyperparameters like the learning rate and batch size? Could the instability observed with GA be tackled with careful hyperparameter tuning specific to that method?

**Relation To Broader Scientific Literature:**

The paper is well-written and establishes itself well in the context of NN solvers for ab-intio methods for the Schrödinger equation, symmetrization in machine learning and VMC methods.

**Theoretical Claims:**

As a reviewer whose expertise lies in physics-informed ML for DFT and not specifically in VMC and associated ML methods, I have reviewed the statements of the theoretical results but have not rigorously checked the correctness of the proofs provided in the appendices. They seem plausible and the authors strengthen the claims with empirical results and have detailed proofs in appendices.

---

> ### Author Rebuttal · Authors · 2025-03-30
>
> We thank the reviewer for the positive and helpful feedback. We address the questions below.
>
> ---
> 1) **Relevance to other areas of physics simulation with similar symmetry constraints arise.** We agree that this would add value and is a very interesting avenue of future work. **In the revision, we will have a discussion and limitation section to discuss the relevance and comparisons with known cases of symmetries in other setups.** We do note that, as the other reviewers have also brought up comparisons with symmetrisation approaches in VMC for non-solid contexts (see response 1.2 to sZMY), we will prioritise those comparisons first and, if space permits, include a brief discussion on implications for other physics setups.
> ---
> 2) **Interest for the ICML community.**  We believe the general message of this paper, especially the part regarding the surprising effects of DA and GA in training, **is of broad interest to the ML community even beyond a physics setup**. While symmetry has become the staple in highly structured problems such as those in AI4Science, the observed benefits of symmetry in practice are often entangled with architectural changes, optimizer changes and hyperparamter choices. **Recent findings in protein structures [1] and atomic potential [2] showed that symmetries may actually be unnecessary for performance improvement**, calling into question how much symmetry helps with performance versus other factors e.g. hyperparameters. Our work can be viewed as **a first step towards this question** in the specific context of VMC for solids, as we perform **a strict apples-to-apples comparison on a fixed architecture with fixed optimizer and hyperparameters** and only vary the symmetrization methods. The computational-statistical tradeoffs we see for DA and GA are in fact applicable to any ML setups where sampling is performed in between gradient updates; one non-physics example in ML would be the contrastive divergence algorithm for training energy-based models (see e.g. [3] and [4]). **We will make sure this general applicability is mentioned in the revision.**
> ---
> 3) **Sensitivity of the results, e.g.~the observed instability, to the choice of architecture and hyperparameters.** We thank the reviewer for the interesting question. For varying batch sizes, we have included a preliminary ablation test in Appendix B.4. For the learning rate, we used the default setup from DeepSolid to ensure an apples-to-apples comparison across all setups. In view of our theoretical results, we believe that the instability findings and the computational-statistical tradeoffs are independent of these hyperparameters. **For architectures, we only used DeepSolid since it is the only architecture we know that performs well for VMC with infinite periodic solids, and because we want to perform an apples-to-apples comparison to an architecture that is known to do well in these problems.** We do conjecture that the effects of symmetrization may change in the context of other architectures. One example is the situation where, the system of interest possesses simpler symmetries and where more efficient symmetrizations exist (discussed in more detail in response 1.2 to reviewer szMY). Meanwhile, we agree that it is interesting to explore whether there exists a way to tweak DeepSolid in the ``most optimal way" for DA or GA such that it outperforms PA. Although we feel that this is out of scope for the current paper, we do hope that our findings pave the way for these investigations: This is both by our discussions on the computational and statistical costs of DA and GA that one needs to be aware of when benchmarking the performances, and by offering tools such as the visualization method for understanding symmetry improvements.
> ---
> We hope these answer your questions. If our responses are helpful, we would be grateful if you would consider raising your score.
>
> ---
>
> **References used for this response and other responses**
>
> [1] Abramson, Josh, et al. "Accurate structure prediction of biomolecular interactions with AlphaFold 3." Nature 2024
>
> [2] Qu, E, and Aditi K. "The importance of being scalable: Improving the speed and accuracy of neural network interatomic potentials across chemical domains." NeurIPS 2024
>
> [3] Hinton, G. E. "Training products of experts by minimizing contrastive divergence." Neural computation 2002
>
> [4] Du, Y, et al. "Improved contrastive divergence training of energy based models." ICML 2021
>
> [5] Chen, S., Edgar D., and Jane H. Lee. "A group-theoretic framework for data augmentation." JMLR 2020
>
> [6] Lyle, C., et al. "On the benefits of invariance in neural networks." arXiv:2005.00178
>
> [7] Benton, G., et al. "Learning invariances in neural networks from training data." NeurIPS 2020
>
> [8] Yang, Jianke, et al. "Generative adversarial symmetry discovery." ICML 2023
>
> [9] Balestriero, R, Leon B, and Yann L. "The effects of regularization and data augmentation are class dependent." NeurIPS 2022

---

> > ### Comment · Reviewer_mfuU · 2025-04-07
> >
> > Thank you for the detailed rebuttal. Your clarifications on the broader context has convinced me about it's relevance to ICML. I am increasing my score accordingly and look forward to the revised manuscript.

---

### Decision · Program_Chairs · 2025-05-01

**Decision:**

Accept (poster)

**Comment:**

The authors compare different strategies for incorporating discrete symmetries of the sort that exist in materials into neural network ansatzes. They somewhat surprisingly find that, while explicitly symmetrizing during training can lead to instabilities, training an ansatz which does not respect discrete symmetries and then symmetrizing at inference time actually improves performance relative to the trained nonsymmetric ansatz. The reviews were generally positive. The paper could be improved with the inclusion of experiments on larger systems - only quite small systems were studied (LiH, graphene) and only a single unit cell, without twist averaging, so the results cannot be compared to experiment. Nevertheless, it's a surprising finding that could be useful for evaluating models trained on larger systems.